# A Quantum Circuit-Based Compression Perspective for Parameter-Efficient Learning

**Chen-Yu Liu**[1,2]   **Chao-Han Huck Yang**[3]   **Hsi-Sheng Goan**[1,4,5,6]   **Min-Hsiu Hsieh**[2]

[1] Graduate Institute of Applied Physics, National Taiwan University, Taipei, Taiwan

[2] Hon Hai (Foxconn) Research Institute, Taipei, Taiwan

[3] Georgia Institute of Technology, USA

[4] Department of Physics and Center for Theoretical Physics, National Taiwan University, Taipei, Taiwan

[5] Center for Quantum Science and Engineering, National Taiwan University, Taipei, Taiwan

[6] Physics Division, National Center for Theoretical Sciences, Taipei, Taiwan

{d10245003@g,goan@phys}.ntu.edu.tw,
huckiyang@gatech.edu, min-hsiu.hsieh@foxconn.com

## Abstract

Quantum-centric supercomputing presents a compelling framework for large-scale hybrid quantum-classical tasks. Although quantum machine learning (QML) offers theoretical benefits in various applications, challenges such as large-size data encoding in the input stage and the reliance on quantum resources in the inference stage limit its practicality for tasks like fine-tuning large language models (LLMs). Quantum parameter generation, a novel approach of QML, addresses these limitations by using quantum neural networks (QNNs) to generate classical model weights (parameters) exclusively during training, thereby decoupling inference from quantum hardware. In this work, we introduce Quantum Parameter Adaptation (QPA) in the framework of quantum parameter generation, which integrates QNNs with a classical multi-layer perceptron mapping model to generate parameters for fine-tuning methods. Using Gemma-2 and GPT-2 as case studies, QPA demonstrates significant parameter reduction for parameter-efficient fine-tuning methods, such as Low-Rank Adaptation (LoRA), while maintaining comparable or improved performance in text generation tasks. Specifically, QPA reduces the number of parameters to $52.06\%$ of the original LoRA for GPT-2 with a slight performance gain of $0.75\%$, and to $16.84\%$ for Gemma-2, with a marginal performance improvement of $0.07\%$. These results highlight QPA's ability to achieve efficient parameter reduction without sacrificing performance in the quantum parameter generation framework. This work showcases the potential of quantum-enhanced parameter reduction, offering a scalable quantum-classical solution for fine-tuning LLMs while preserving the feasibility of inference on classical hardware.

## 1 Introduction

Decomposing complex problems into components suited for classical or quantum computing allows for more efficient problem-solving. Classical systems are well-suited for tasks like data processing, while quantum computing shows potential in optimization and exploring large state spaces. Quantum-centric supercomputing (Bravyi et al., 2022; Gambetta, 2022) combines these strengths, offering scalable solutions for challenging problems. This hybrid approach holds promise in quantum machine learning (QML), with the potential to support the training and fine-tuning of large models. Conventional QML approaches employ parameterized quantum circuits (PQCs) as quantum neural networks (QNNs) (Chen et al., 2020), where data is input through specific data encoding methods (Pérez-Salinas et al., 2020; Schuld et al., 2021). The updates to QNN parameters during the training process are computed on the classical side, creating a hybrid quantum-classical computing framework (Mari et al., 2020). When scaled up, quantum-centric supercomputing offers a promising paradigm that could reshape the landscape of computational science, particularly in the current Noisy Intermediate-Scale Quantum (NISQ) era (Preskill, 2018). Classical neural networks

(NNs) can also be combined with QNNs, serving as either pre-processing or post-processing layers (Mari et al., 2020; Liu et al., 2021). While both empirical and theoretical results suggest that QML can offer improvements in specific applications (Cerezo et al., 2022; Huang et al., 2022; Biamonte et al., 2017; Caro et al., 2022; Huang et al., 2021), significant challenges remain—particularly in data encoding for large datasets. For instance, gate-angle encoding can result in quantum circuits that are either too deep or require an impractical number of qubits. Another major obstacle to the practicality of QML is the need for a quantum computer during the model inference stage.

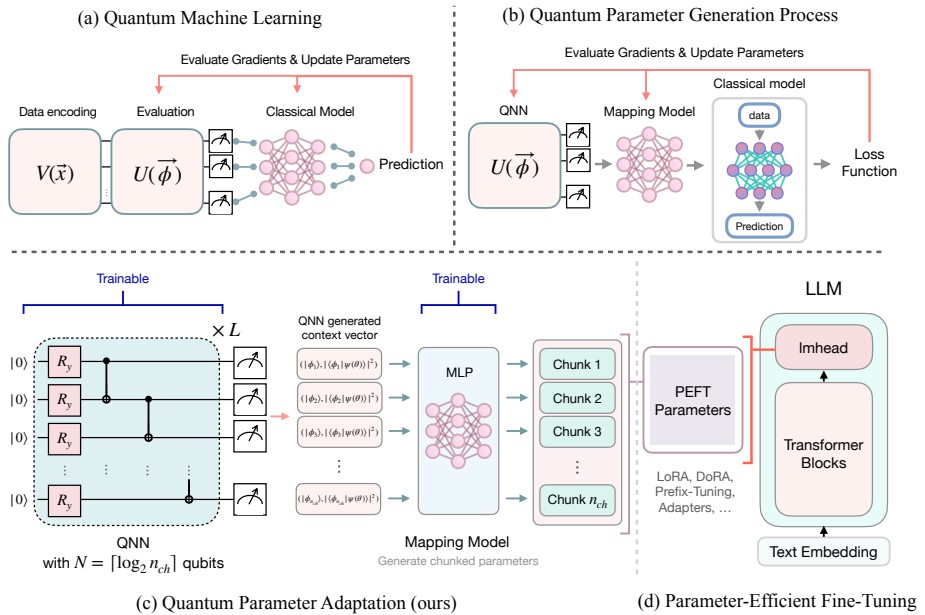

Figure 1: Overview of (a) Quantum Machine Learning (Mari et al., 2020; Mitarai et al., 2018), (b) Quantum Parameter Generation (Liu et al., 2024b), (c) our proposed Quantum Parameter Adaptation (QPA), and (d) Parameter-Efficient Fine-Tuning methods (Houlsby et al., 2019).

One such proposal to address both the data encoding challenge and the requirement for quantum hardware during the inference stage is quantum parameter generation (Sidhu & Kok, 2020; Liu et al., 2024b). Instead of using a QNN to interact directly with the data, this approach leverages QNNs to "generate" the weights of a target classical NN model during the training process. This quantum parameter generation method: 1) keeps the data input process entirely within the classical model, 2) eliminates the exorbitant need for quantum computing hardware during inference, as the trained model is purely classical, and 3) reduces the number of training parameters on a *polylogarithmic* scale by mapping quantum state basis to the target NN parameters, assuming the QNN is constructed with a polynomial number of layers relative to the qubit count. Further details will be provided in the following sections.

Current studies (Liu et al., 2024c; Lin et al., 2024) of quantum parameter generation have primarily focused on small-scale models with fewer than one million parameters, where the largest model is up to 0.28M. To more clearly demonstrate the practicality of this quantum approach, the size of the models or problems investigated should be scaled to more realistic levels, extending beyond basic tasks such as handwritten digit classification and simplified reinforcement learning environments.

As model sizes grow, both in quantum and classical machine learning (ML), scaling becomes a key computation challenge. This is especially evident in the rise of large language models (LLMs), which have become foundational in natural language processing tasks. However, training and fine-tuning LLMs—such as GPT-2 (Radford et al., 2019), GPT-3 (Brown, 2020), and the more recent Gemma-2 (Team et al., 2024)—presents its own set of challenges. The immense number of parameters in these models requires vast computational resources, making conventional fine-tuning

approaches computationally expensive and impractical for many use cases. To address these issues, parameter-efficient fine-tuning methods (PEFT) like Low-Rank Adaptation (LoRA) (Hu et al., 2021), Weight-Decomposed Low-Rank Adaptation (DoRA) (Liu et al., 2024d), Prefix-Tuning (PT) (Li & Liang, 2021; Yang et al., 2021b), and adapters have been proposed (Houlsby et al., 2019; Lin et al., 2020). These methods aim to reduce the number of trainable parameters while maintaining model performance, yet the challenge of balancing efficiency and effectiveness remains a critical area of research.

We draw inspiration from both quantum parameter generation and PEFT methods for LLMs, which demonstrate that PQCs can serve as parameter generators for machine learning models, and changes to LLM weights can be captured by low-rank or smaller-sized representations (models). We suggest that the training parameters in PEFT methods can be generated using a combination of QNNs and a classical multi-layer perceptron (MLP) mapping model, further reducing the parameter count. This leads to our introduction of the **Q**uantum **P**arameter **A**daptation (QPA) method. QPA enables additional parameter reduction during fine-tuning by leveraging the quantum parameter generation method to generate PEFT parameters, as shown in Fig. 1. The figure also provides an overview of QML, quantum parameter generation, and PEFT for comparison, clearly illustrating how QPA generates the PEFT parameters. In experiments with Gemma-2 (Team et al., 2024) and GPT-2 (Radford et al., 2019), we demonstrate that QPA reduces the number of parameters in methods such as LoRA, DoRA, PT, and Feed-Forward Adapters (FFAs) by nearly an order of magnitude (e.g., $10^6 \rightarrow 10^5$), while achieving better or comparable perplexity in text generation tasks. QPA marks **the first example** of quantum computing applied to fine-tuning classical LLMs at a practical scale, while enabling inference to be performed entirely on classical hardware. Further theoretical analysis of QPA's convergence behavior, as well as its trainability and learnability properties, is left for future work.

The main contributions of QPA include:

- **Efficient adaptation through quantum parameter generation**: QPA utilizes quantum parameters from the PQC, which scale in proportion to the number of qubits $N$, to generate parameters that scale with the Hilbert space size $2^N$. When applied to tunable parameters in PEFT tasks, this property enables highly efficient adaptation by generating parameters through quantum methods.

- **Practical application of QML in LLMs**: By incorporating quantum parameter generation, our approach to combining QML with LLMs is more practical compared to conventional QML methods (i.e., Quantum Transformer (Di Sipio et al., 2022)). The data-encoding issues associated with conventional QML (Yang et al., 2021a) have been eliminated, and the inference stage no longer requires quantum hardware. This is particularly important for tasks that demand short response times, as reliance on remote quantum computers could introduce delays due to queuing and increased costs.

- **Scaling up the size of existing quantum parameter generation studies**: In this study, we scale up the application of quantum parameter generation by fine-tuning up to the linear layer of the Gemma-2 (2B) model, where the target layer consists of $0.52B$ parameters. This is approximately 1785 times larger than the largest previously studied model ($0.28M$), significantly pushing the boundaries of how QML can be applied.

- **Flexibility, applicability, and efficiency in classical fine-tuning methods**: We demonstrate that quantum parameter generation is not only capable of generating classical NN parameters, but is also applicable to a wide-range of popular parameter-tuning tasks. These tasks include tuning parameters in methods such as LoRA, DoRA, PT, and FFA, all while achieving comparable or better perplexity in text generation tasks with reduced parameter requirements.

## 2 RELATED WORKS

**Parameter-Efficient Fine-Tuning (PEFT) Methods.** To address the challenge of fine-tuning LLMs, PEFT methods aim to reduce the number of trainable parameters while maintaining or even improving model performance. Key PEFT approaches include LoRA (Hu et al., 2021) and DoRA (Liu et al., 2024d), which assume that the changes in model weights during fine-tuning lie in a low-rank subspace. Instead of updating the full weight matrices, these methods add small, train-

able low-rank decomposition matrices to the model's weights, effectively reducing the number of trainable parameters while capturing the essential adjustments. Another approach involves adapters, which are small feed-forward layers inserted between existing layers of the neural network (Houlsby et al., 2019; Lin et al., 2020). Only these adapter layers are fine-tuned, leaving the rest of the model unchanged, thus significantly reducing the training overhead. PT (Li & Liang, 2021; Yang et al., 2021b) takes a different approach by introducing tunable prefix vectors that are prepended to the input or hidden states at each layer. During fine-tuning, only these prefix vectors are updated, while the core model parameters remain frozen. These techniques optimize fine-tuning efficiency by focusing on smaller, task-specific modifications.

**From Quantum Circuit Learning to Large-Scale Representation Modeling.** QML leverages quantum properties such as superposition and entanglement, presenting a theoretically promising approach to accelerate the training and learning process (Du et al., 2021a; Schuld & Killoran, 2019; Farhi & Neven, 2018). However, the process of encoding classical data into quantum states increases the depth of quantum circuits, further adding to the depth of QNNs (Pérez-Salinas et al., 2020; Schuld et al., 2021). We refer to QML approaches that directly interact with data as "conventional QML," distinct from the quantum parameter generation approach discussed later. There have been proposals suggesting that the GPT model could be integrated into the quantum computing paradigm (Liao & Ferrie, 2024), where the transformer architecture is implemented using quantum circuits (Gao et al., 2023; Khatri et al., 2024). In such case, the input data is encoded into quantum states through amplitude encoding methods. Consequently, these QML methods require access to quantum computers during the inference stage.

**Training Classical Neural Networks via Quantum Computing.** Leveraging quantum computing to train classical NNs offers a promising approach where the resulting model remains entirely classical, effectively addressing the challenges of data encoding and the reliance on quantum hardware during inference. One example involves utilizing quantum walks as a search process to optimize the parameters of classical NNs (de Souza et al., 2021). Another method proposes employing a quantum hypernetwork to train a classical binary NN (Carrasquilla et al., 2023). However, the practical utility of this approach is constrained by its ability to train only binary NNs, limiting its applicability in more general settings.

## 3 Quantum Parameter Generation Based Efficient Adaptation

In LoRA and DoRA, one primary assumption is that weight updates can be represented by low-rank matrices. However, these methods often lack flexibility in exploring parameter settings between integer ranks $r$ and $r + 1$, leaving the intermediate values unexamined. A method capable of investigating the parameter space between $r$ and $r + 1$ could potentially enhance performance by enabling more precise, "fine-grained" adjustments. This limitation also applies to other PEFT methods with rigid configurations, such as PT, where the number of trainable parameters is strongly tied to the input size.

Inspired by this idea and combined with quantum circuit based compression, which uses PQC and mapping model to train the target NN with fewer parameters, we hypothesize that using PQC-based QNN and mapping model as a parameter generator can take advantage of the fact that only a small number of QNN parameters are required to control the measurement probabilities, governed by the dimension of the Hilbert space. In other words, we propose that the high-dimensional Hilbert space facilitates an efficient representation for adaptation. This could lead to a more detailed and precise exploration of the representation space in PEFT methods.

### 3.1 Quantum Circuit based Model Parameter Generation.

We consider a parameter generation process that provides a different approach from conventional QML. Consider a target NN model with parameters $\boldsymbol{a} = (a_1, a_2, \ldots, a_m)$, where $m$ is the number of parameters. A PQC with $N = \lceil \log_2 m \rceil$ qubits and $L$ layers is constructed using a circuit ansatz, represented as:

$$|\psi(\boldsymbol{\theta})\rangle = \left( \prod_{i=1}^{N-1} \text{CNOT}^{i,i+1} \prod_{j=1}^{N} R_Y^j(\theta_j^{(L)}) \right)^L |0\rangle^{\otimes N},$$  (1)

where the single-qubit rotation gate $R_Y^j$ is associated with the tunable parameter $\theta_j^{(L)}$, with qubit index $j$ and layer index $L$, and CNOT represents the two-qubit controlled-NOT gate. With a Hilbert space of size $2^N$, where $2^N \geq m$, this PQC produces $2^N$ distinct measurement probabilities, $|\langle \phi_i | \psi(\boldsymbol{\theta}) \rangle|^2 \in [0, 1]$ for $i \in \{1, 2, \ldots, 2^N\}$. The parameter size of $\boldsymbol{\theta}$ depends on $N$ and $L$, where $L$ is a hyperparameter similar to those in classical ML. Typically, $L$ scales proportionally to the number of qubits, either $O(N)$ or in some cases $O(N^2)$ (Cerezo et al., 2021; Sim et al., 2019; Benedetti et al., 2019), though it can be generalized to a looser polynomial scale, $O(poly(N))$. Thus, with polynomial layers in the PQC, we can generate $2^{\lceil \log_2 m \rceil} \geq m$ parameters (probabilities) using $O(polylog(m))$ PQC parameters.

At this stage, the measurement probabilities are values between 0 and 1. To map these probabilities to the target parameters $\boldsymbol{a} \in \mathbb{R}^m$, we employ a MLP mapping model $G$ with tunable parameters $\boldsymbol{b}$. The input to $G$ is the binary representation of the basis (of length $N$) and the corresponding measurement probability, such that

$$G_{\boldsymbol{b}}(|\phi_i\rangle, |\langle \phi_i | \psi(\boldsymbol{\theta}) \rangle|^2) = a_i, \quad \forall i \in \{1, 2, \ldots, m\}. \tag{2}$$

Here, only the first $m$ basis states are used to cover all parameters in the target NN. Since the input size of $G_{\boldsymbol{b}}$ is $N+1$, the size of $\boldsymbol{b}$ can also be controlled at a scale of $O(polylog(M))$. Consequently, $\boldsymbol{a}$ is generated from the output of the PQC and the mapping model $G_{\boldsymbol{b}}$. By tuning $\boldsymbol{\theta}$ and $\boldsymbol{b}$, we effectively influence the value of the loss function $\mathcal{L}$, which is evaluated by the target NN for a given task.

*Gradient Estimation of Quantum Circuit Compressed Parameters.* The target NN parameters $\boldsymbol{a}$ are generated through the use of a PQC coupled with a mapping model. The quantum-dependent parameters, represented as $(\boldsymbol{\theta}, \boldsymbol{b})$, impact the target NN parameters via the quantum state preparation and measurement steps. The gradient of the loss function, reflecting the influence of quantum parameters, is expressed as:

$$\nabla_{\boldsymbol{\theta}, \boldsymbol{b}} \mathcal{L} = \left( \frac{\partial \boldsymbol{a}}{\partial (\boldsymbol{\theta}, \boldsymbol{b})} \right)^T \cdot \nabla_{\boldsymbol{a}} \mathcal{L}. \tag{3}$$

In this expression, $\frac{\partial \boldsymbol{a}}{\partial (\boldsymbol{\theta}, \boldsymbol{b})}$ denotes the Jacobian matrix, which describes how sensitive the classical parameters $\boldsymbol{a}$ are to changes in the quantum parameters $(\boldsymbol{\theta}, \boldsymbol{b})$. This provides a high-level overview of the gradient in an exact quantum state simulation. In practical applications involving real quantum computers, the gradient calculation must account for the parameter shift rule and its variants (Mitarai et al., 2018; Schuld et al., 2019).

*Parameter Update of Quantum Circuit Compressed Parameters.* The learning rate $\eta$ is a critical factor, particularly due to the complex dynamics introduced by the quantum-classical interface. The update for the quantum parameters is defined as:

$$\boldsymbol{\theta}_{t+1}, \boldsymbol{b}_{t+1} = \boldsymbol{\theta}_t, \boldsymbol{b}_t + \eta \nabla_{\boldsymbol{\theta}, \boldsymbol{b}} \mathcal{L}. \tag{4}$$

This rule ensures that the quantum parameters are updated to optimize the performance of the target NN.

Using the gradient computation and parameter update rule, the parameter generation process of QPA has been applied to image classification with convolutional neural networks (CNNs) (Liu et al., 2024b;a; Liu & Chen, 2024), flood prediction (time series) with long short-term memory (LSTM) models (Lin et al., 2024), and policy gradient reinforcement learning in CartPole and MiniGrid environments (Liu et al., 2024c). These applications demonstrated a reduction in training parameters while maintaining similar performance.

Table 1: Configuration of the mapping model $\tilde{G}_{\boldsymbol{b}}$, with $N$ representing the number of qubits for each task.

| Hyperparameter | Meaning | Value |
|---|---|---|
| Input size | Input of the mapping model $(|\phi_i\rangle, |\langle \phi_i | \psi(\boldsymbol{\theta}) \rangle|^2)$ | $N+1$ |
| Hidden dimension | Main structure of the MLP mapping model | $[32, 64, 128, 128, 64, 32, n_{mlp}]$ |

## 3.2 BATCHED PARAMETER GENERATION OF QUANTUM PARAMETER ADAPTATION (QPA)

If the target NN model consists of $m$ parameters, the required number of qubits is $N = \lceil \log_2 m \rceil$. For instance, scaling this up to $m = 10^9$ (one billion) parameters would require $N = 30$ qubits. Although a quantum system with 30 qubits is feasible on today's hardware and in classical simulations, the GPU memory requirements (around 16 GB) and evaluation time (several seconds per circuit) (Google, 2024) make it impractical for ML tasks that demand extensive repetitions within a reasonable time frame.

In our approach, the $m$ parameters (following the above syntax) of the target NN model are split into $n_{ch}$ chunks, with each chunk containing $n_{mlp}$ parameters, such that $n_{ch} = \lceil m/n_{mlp} \rceil$. In other words, the qubit count can be reduced by generating more than one parameter of the target NN per quantum basis.

The mapping model, now represented as $\tilde{G}_{\boldsymbol{b}}$, takes as input $|\phi_i\rangle$ and $|\langle\phi_i|\psi(\boldsymbol{\theta})\rangle|^2$, and outputs a batch of $n_{mlp}$ parameters $\boldsymbol{a}$ for each chunk:

$$\boldsymbol{a} = (\tilde{a}_1, \tilde{a}_2, \ldots, \tilde{a}_{n_{ch}}), \tag{5}$$

$$\tilde{G}_{\boldsymbol{b}}(|\phi_i\rangle, |\langle\phi_i|\psi(\boldsymbol{\theta})\rangle|^2) = \tilde{a}_i, \quad \forall i \in \{1, 2, \ldots, n_{ch}\}, \tag{6}$$

$$\tilde{a}_i = (a_{i,1}, a_{i,2}, \ldots, a_{i,j}), \quad \forall j \in \{1, 2, \ldots, n_{mlp}\}. \tag{7}$$

This is achieved by using a decoder-like structure in the MLP within the mapping model $\tilde{G}_{\boldsymbol{b}}$, where the output size is expanded from 1 to $n_{mlp}$, the detailed configuration is shown in Table. 1. As a result, the number of qubits required is reduced from $N = \lceil \log_2 m \rceil$ to

$$N = \lceil \log_2 n_{ch} \rceil = \lceil \log_2 \left( \lceil \frac{m}{n_{mlp}} \rceil \right) \rceil. \tag{8}$$

This adjustment effectively reduces the qubit requirement by approximately $\lceil \log_2 n_{mlp} \rceil$ qubits compared to the original quantum parameter generation. The original method can be viewed as a special case where $n_{mlp} = 1$. Although this reduction in qubit count increases the number of parameters in the mapping model (due to the expanded output size), it offers a significant memory reduction, as the memory needed to store the quantum state decreases by a factor of $1/n_{mlp}$.

For example, with $m = 10^9$ and $n_{mlp} = 1024$, the qubit usage can be reduced to $N = \lceil \log_2 \left( \lceil \frac{10^9}{1024} \rceil \right) \rceil = 20$. This 33% of qubit saving reduces the memory required to store the quantum state to just $1/1024$ of what is needed for 30 qubits, and speedup the classical simulation as well.

## 3.3 FROM MODEL TUNING TO GENERAL PARAMETER-TUNING TASK

In QPA, we consider the tuning target of quantum parameter generation from the full model parameters of a target NN to the parameters of a PEFT method, enabling significantly larger tasks to be handled. Additionally, with the help of batch parameter generation, the scale of the target task is further amplified. Following the previous notation, the parameter $\boldsymbol{a}$ now represents the parameters of a PEFT method.

Taking LoRA as an example, for a pre-trained weight matrix $W_0 \in \mathbb{R}^{d \times k}$, the low-rank decomposition of the update is $W_0 + \Delta W = W_0 + BA$, with $B \in \mathbb{R}^{d \times r}$, $A \in \mathbb{R}^{r \times k}$, and $r \ll \min(d, k)$. In this case, QPA generates these two low-rank matrices using Eq. 6 and Eq. 7, where $\boldsymbol{a}$ represents the elements of $A$ and $B$, and the required qubit count is:

$$N = \lceil \log_2 \left( \lceil \frac{r(d + k)}{n_{mlp}} \rceil \right) \rceil \quad \text{(LoRA case)}. \tag{9}$$

The tuning of QPA can then follow the gradient evaluation and update rules in Eq. 3 and Eq. 4. A similar situation applies to DoRA, where there are additional $k$ parameters in the magnitude vector to be tuned. In this case, the required qubit count is:

$$N = \lceil \log_2 \left( \lceil \frac{r(d + k) + k}{n_{mlp}} \rceil \right) \rceil \quad \text{(DoRA case)}. \tag{10}$$

These examples can be generalized to other parameter-tuning tasks. The schematic of QPA is shown in Fig. 1, where the PEFT method is applied to only one layer in the LLM, as we will discuss in Sec. 4, which describes the experiment. For scenarios where gradient-based tuning is not supported, other non-gradient-based optimizers (e.g. Nelder-Mead and COBYLA) may be required for the parameter update process.

## 4 EMPIRICAL EXPERIMENTS

Our objective is to assess the effectiveness of QPA, as outlined in Sec. 3, to determine whether the proposed QPA can effectively reduce the number of parameters while maintaining or surpassing the performance of existing PEFT methods. This evaluation is based on one hypothesis that the high-dimensional Hilbert space enables efficient representation for adaptation. The experiment is conducted using quantum circuit simulation via PyTorch and TorchQuantum (Wang et al., 2022). At this stage, noise effects on the quantum system are ignored, and the quantum state amplitudes (probabilities) are obtained exactly. A discussion on the impact of finite measurement shots and noise is provided in Appendix G. We assess the text generation perplexity of Gemma-2 (2B) and GPT-2 (80M), fine-tuned on the WikiText-2 dataset, using several well-known PEFT methods, including LoRA, DoRA, PT, and Feed-Forward Adapter (FFA). Gemma-2 was selected as one of the most recent LLMs[1], offering competitive performance relative to models such as Phi-2-2B (Javaheripi et al., 2023), LLaMA2-7B (Touvron et al., 2023), and Mistral-7B (Jiang et al., 2023). Additionally, we assess GPT-2 XL (1.5B) with QPA, with further discussion provided in Appendix A.1. For comparison, we apply QPA to generate the parameters for these PEFT methods following the procedure outlined in Sec. 3.3. The full hyperparameter configurations are provided in Appendix C, while additional results on a different dataset are presented in Appendix E. To isolate the effects of QPA, we simplify the PEFT setup by freezing all layers of Gemma-2 and GPT-2, and fine-tuning only the final linear layer, commonly referred to as the "lmhead." This layer comprises 38.59M parameters in GPT-2 and 0.52B parameters in Gemma-2. The QNN repetition $L$ is fixed at 8 for all cases, except those discussed in the "Effect of Deeper QNN" paragraph.

### 4.1 PERFORMANCE OF QPA

**Low-rank adaptation methods with QPA.** First, using LoRA and DoRA as baselines, QPA is applied to generate the low-rank matrices for these methods, with an additional magnitude vector generated for DoRA. In this experiment, the rank of the low-rank matrices to be generated is fixed at 4 (other results could be found at Appendix D), and various chunk sizes $n_{mlp}$ are investigated to adjust the number of trainable parameters. Specifically, $n_{mlp} \in \{256, 512, 1024, 2048, 4096, 8192\}$ for GPT-2 with LoRA, $n_{mlp} \in \{512, 1024, 2048, 4096, 8192\}$ for GPT-2 with DoRA, $n_{mlp} \in \{1024, 4096, 8192, 16258, 32768, 65536\}$ for Gemma-2 with LoRA, and $n_{mlp} \in \{512, 1024, 2048, 4096, 8192\}$. For comparison, the results for LoRA and DoRA are obtained by varying the rank $r \in \{1, 2, 4, 8, 16, 32\}$.

In the results, as illustrated in Fig. 2, QPA consistently outperforms LoRA and DoRA across various parameter configurations. For GPT-2, QPA achieves lower testing perplexity compared to LoRA, particularly in configurations with fewer trainable parameters, demonstrating its efficiency. Even with larger parameter counts, QPA maintains comparable or better performance than LoRA. The best result in terms of parameter reduction and improved perplexity is achieved with 106264 trainable parameters and a perplexity of 1.583 for QPA, while LoRA requires 204100 trainable parameters and achieves a perplexity of 1.595. This indicates that QPA reduces the trainable parameters to 52.06% while delivering a 0.75% improvement in performance. Similarly, for Gemma-2, QPA demonstrates significant improvements over LoRA in the lower parameter regime, with the gap narrowing as the parameter count increases, yet QPA consistently maintains better perplexity overall. The best result here is 173,888 trainable parameters with a perplexity of 1.417 for QPA, compared to 1,032,192 trainable parameters with a perplexity of 1.418 for LoRA. This corresponds to a reduction to 16.84% in trainable parameters with a 0.07% performance improvement. When comparing QPA to DoRA, the results initially show QPA having higher perplexity with fewer parameters for both GPT-2 and Gemma-2. However, as the number of trainable parameters increases, QPA sur-

---

[1]We selected the deployed Gemma-2 version released on August 8th, 2024: `https://huggingface.co/google/gemma-2-2b`.

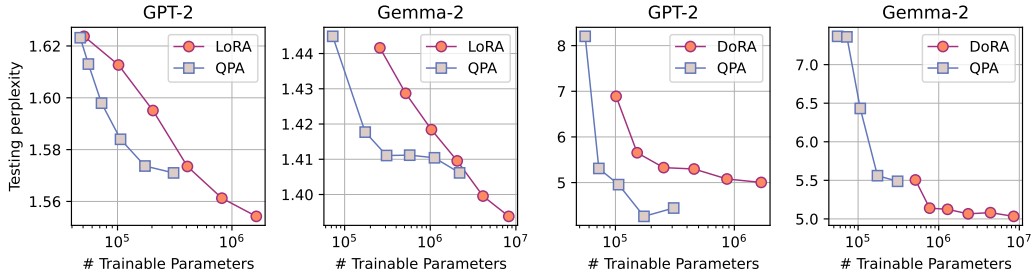

Figure 2: Testing perplexity of GPT-2 and Gemma-2 models compared to the number of trainable parameters for LoRA, DoRA, and QPA on the WikiText-2 dataset.

Table 2: Training parameters and testing perplexity for GPT-2 and Gemma-2 using PEFT and QPA methods, with emphasis on configurations achieving the most significant parameter reductions. Complete results are provided in Fig. 2 and Fig. 3.

| PEFT Method | GPT-2 | | Gemma-2 | |
|---|---|---|---|---|
| | # Params (%) | PPL | # Params (%) | PPL |
| LoRA | 0.52 | 1.595 | 0.19 | 1.418 |
| QPA LoRA (Ours) | **0.27** | **1.583** | **0.03** | **1.417** |
| DoRA | 4.36 | 5.003 | 0.09 | 5.504 |
| QPA DoRA (Ours) | **0.27** | **4.955** | **0.05** | **5.487** |
| PT | 1.01 | 2.225 | 0.20 | 1.530 |
| QPA PT (Ours) | **0.18** | **2.327** | **0.01** | **1.540** |
| FFA | 0.76 | 1.763 | 0.40 | 1.439 |
| QPA FFA (Ours) | **0.18** | **1.689** | **0.01** | **1.507** |

passes DoRA, ultimately achieving comparable or better results. The result with best parameter reduction are concluded in Table. 4.1, with the parameter ratio calculated relative to the number of parameters in the target layer. In the case of Gemma-2, the performance gap between QPA and DoRA also narrows as the number of trainable parameters increases, with QPA remaining competitive across all scales. While the magnitude of these performance improvements may appear small, the key takeaway is that QPA can significantly reduce the number of trainable parameters in PEFT methods without incurring significant loss in performance.

**QPA on Prefix-Tuning and Feed-forward adapter.** In the context of PT, QPA is applied to the tunable prefix vector that is prepended to the input of the target linear layer, where the length of this vector is traditionally fixed to match the input size of the layer. QPA offers a more flexible approach by allowing different parameter configurations, enabling exploration beyond the conventional constraints of input length. The QPA setup uses various chunk sizes $n_{mlp} \in \{256, 512, 1024, 2048, 4096, 8192\}$ for GPT-2 and $n_{mlp} \in \{1024, 2048, 4096, 8192, 16258, 32768\}$ for Gemma-2, with an additional $n_{mlp} = 65536$ for QPA-FFA case.

As shown in Fig. 3 and Table. 4.1, for GPT-2, QPA does not outperform PT at lower parameter counts but shows potential for extending the parameter space and achieving better results as the number of parameters increases. In the scenario with the most parameter reduction, QPA achieves $72,552$ trainable parameters and a perplexity of 2.327, compared to PT with $393,216$ trainable parameters and a perplexity of 2.225. This indicates that QPA reduces the number of parameters to $18.45\%$, at the cost of a $4.38\%$ performance loss.

For Gemma-2, QPA initially performs on par with PT but surpasses it as the number of trainable parameters increases, leading to superior performance at higher parameter scales. This flexibility in parameter tuning afforded by QPA allows for a broader exploration of the parameter space, potentially resulting in better outcomes than traditional PT. By inserting small feed-forward layers before the target linear layer, QPA generates the parameters for the FFA. For GPT-2, QPA initially outper-

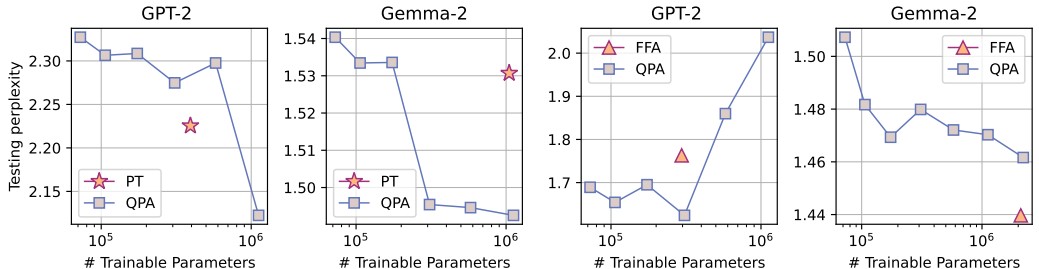

Figure 3: Testing perplexity of GPT-2 and Gemma-2 models compared to the number of trainable parameters for prefix-tuning (PT), feed-forward adapter (FFA), and QPA on the WikiText-2 dataset.

forms FFA at lower parameter counts, but as the number of parameters increases, QPA's performance deteriorates compared to FFA. In contrast, for Gemma-2, QPA does not outperform the classical FFA approach across the entire parameter range. However, it still demonstrates competitive performance, maintaining comparable testing perplexity even with larger parameter counts. Although QPA does not consistently outperform PT and FFA in terms of testing perplexity, the significant reduction in trainable parameters ($0.20\% \rightarrow 0.01\%$ for PT and $0.40\% \rightarrow 0.01\%$ for FFA on Gemma-2, as shown in Table 4.1) with only a slight performance loss remains a notable and promising result.

## 4.2 EFFECTS OF QPA SETTINGS

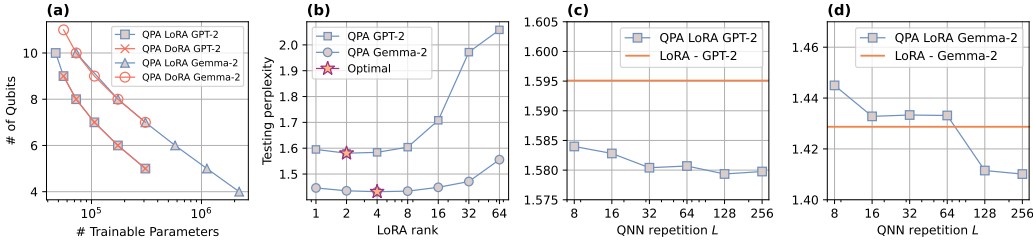

Figure 4: (a) Qubit usage versus the number of trainable parameters for QPA applied to LoRA and DoRA on GPT-2 and Gemma-2 models. (b) The relationship between testing perplexity and LoRA rank for QPA applied to GPT-2 and Gemma-2. (c) and (d) Testing perplexity depending on the QNN repetition $L$ for QPA applied to LoRA on GPT-2 and Gemma-2.

**Trade-off of qubits counts and parameter size**. As described in Sec. 3.2, the qubit usage in QPA follows Eq. 8. Increasing the chunk size $n_{mlp}$ for batched parameter generation increases the total number of trainable parameters in QPA, as $n_{mlp}$ also determines the output size of the MLP mapping model. Consequently, as per Eq. 8, the required qubit count decreases. Fig. 4 (a) illustrates the qubit usage corresponding to the QPA results shown in Fig. 2. The actual qubit usage ranges between 4 and 11 qubits, which are reasonable values for both classical simulations and fault tolerant quantum computers in the foreseeable future [2]. While QPA settings with more qubits tend to reduce the number of trainable parameters in PEFT, this often results in higher perplexities. To achieve optimal performance, $n_{mlp}$ should be carefully tuned to balance the number of trainable parameters and the resulting perplexity.

**Optimal LoRA rank.** As observed in previous LoRA and DoRA studies, increasing the rank of the low-rank matrices does not necessarily lead to better results. In fact, there exists an optimal rank for

---

[2]IBM Quantum announced the achievement of discovering the quantum error correcting code with the property of preserving up to 12 logical qubits using 288 physical qubits through error correction methods (Bravyi et al., 2024): https://www.ibm.com/quantum/blog/nature-qldpc-error-correction.

the best performance. A similar behavior can be seen when applying QPA to low-rank adaptation methods. As shown in Fig. 4 (b), the optimal LoRA ranks (indicated by stars) highlight that QPA achieves its best performance at a LoRA rank of 2 for GPT-2 and 4 for Gemma-2, with Gemma-2 consistently maintaining lower perplexity than GPT-2 across all ranks. The chunk size is fixed at $n_{mlp} = 2048$ for GPT-2 and $n_{mlp} = 8192$ for Gemma-2. As the LoRA rank increases, testing perplexity for GPT-2 rises significantly, while for Gemma-2, it remains relatively stable.

**Effect of Deeper QNN.** In this study, the QNN (quantum circuit) ansatz is shown in Eq. 1, constructed by $R_Y$ and CNOT gates. The performance comparison of different circuit ansatz is presented in Appendix F. The expressiveness of the QNN strongly depends on the number of repetitions $L$ of the corresponding ansatz. A larger number of $L$ results in a deeper QNN, which enhances expressiveness. In Fig. 4 (c) and (d), the testing perplexity results for different QNN repetitions $L$ in QPA applied to LoRA on GPT-2 and Gemma-2 are shown. The orange lines represent the baseline LoRA testing perplexity without QPA, where LoRA rank $= 4$ for GPT-2 and rank $= 2$ for Gemma-2. Fig. 4 (c) shows that for GPT-2, QPA LoRA achieves lower perplexity, which slightly decreases as $L$ increases, with the LoRA rank fixed at 4 and $n_{mlp} = 2048$. Similarly, in Fig. 4 (d), for Gemma-2 (LoRA rank $= 2$ and $n_{mlp} = 8192$), QPA LoRA initially performs comparably to standard LoRA but begins to outperform LoRA when $L$ exceeds 64, with more significant reductions in perplexity observed at higher values of $L$.

In general, when using a universal gate set, such as one containing $R_X$, $R_Y$, $R_Z$, phase shift $P(\varphi)$, and CNOT gates, the Solovay–Kitaev theorem (Dawson & Nielsen, 2005) guarantees that a QNN can approximate any unitary transformation to arbitrary precision. Thus, with sufficient depth (i.e., larger $L$), a QNN constructed from such a gate set can approximate any quantum state, including those that optimally map to the PEFT parameters. As the depth of the QNN increases, it gains the expressivity required to model more complex transformations (Du et al., 2020; Childs, 2017), allowing it to approximate optimal parameter configurations for fine-tuning. This is evident in the reduction of testing perplexity as $L$ grows, illustrating the advantages of deeper QNNs in capturing richer quantum representations. Notably, in practice, good performance can still be attained with a more restricted gate set and practical layer depths. In our study, we employ $R_Y$ and CNOT gates with $L = 8$ in the main experiments, and extend up to $L = O(N^2)$ in this section, demonstrating the scalability of the approach. Notably, an investigation into the gradient variance of quantum circuit parameters is presented in Appendix H. Our findings indicate that the gradient variance does not exhibit the exponential vanishing behavior (commonly referred to as the barren plateau) as qubit usage increases. However, a slight downward trend is observed with increasing $L$.

## 5 CONCLUSION

In conclusion, this work introduces Quantum Parameter Adaptation (QPA) as a new approach for enhancing PEFT methods by leveraging QNNs to generate trainable parameters for LLMs. QPA addresses the critical challenge of reducing the number of parameters required for fine-tuning, which is particularly important for large-scale models. By incorporating QPA into established PEFT methods, such as LoRA, DoRA, PT, and FFA, we demonstrate that QPA *significantly reduces the trainable parameters* (*i.e.*, from $0.40\%$ to $0.01\%$) while maintaining comparable or even improving model performance in text generation tasks. Moreover, the decoupling of near-term available quantum resources (i.e., 4 to 11 qubits) from the inference phase ensures that the quantum benefits are leveraged during training without adding deployment complexities. These results underscore the scalability and efficiency of QPA in fine-tuning LLMs, making it a promising solution for quantum-classical hybrid computational frameworks.

Looking ahead, future research will focus on a more comprehensive theoretical investigation of QPA, particularly in terms of convergence behavior, trainability, and learnability. Expanding QPA's application across a broader range of neural network architectures and tasks beyond text generation will be critical for validating its generalizability and robustness. Furthermore, while we have analyzed QPA using a real quantum computer noise model in Appendix G, conducting experiments on actual quantum hardware will be a crucial step toward developing practical quantum-classical hybrid solutions. These efforts will contribute to establishing QPA as a foundational approach for fine-tuning LLMs in the emerging landscape of quantum-centric supercomputing.

ACKNOWLEDGMENTS

H.-S. Goan acknowledges support from the National Science and Technology Council, Taiwan, under Grants No. NSTC 113-2112-M-002-022-MY3, No. NSTC 113-2119-M-002-021, No. NSTC 114-2119-M-002-017-MY3, from the US Air Force Office of Scientific Research under Award Number FA2386-23-1-4052 and from the National Taiwan University under Grants No. NTU-CC114L895004 and No. NTU-CC-113L891604. H.-S. Goan is also grateful for the support of the "Center for Advanced Computing and Imaging in Biomedicine (NTU-114L900702)" through the Featured Areas Research Center Program within the framework of the Higher Education Sprout Project by the Ministry of Education (MOE), Taiwan, the support of Taiwan Semiconductor Research Institute (TSRI) through the Joint Developed Project (JDP) and the support from the Physics Division, National Center for Theoretical Sciences, Taiwan.

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

# A    APPENDIX: A DISCUSSION ON MODEL COMPRESSION

Model compression is a critical and evolving area of research aimed at enabling efficient training and inference in machine learning models. Common techniques include quantization (Dettmers et al., 2022; 2024; Frantar et al., 2022), pruning (Ma et al., 2023; Sun et al., 2023), and the PEFT methods discussed in the main text. Quantization reduces memory and computational costs by lowering the precision of model weights, while pruning removes less significant weights, thereby reducing the overall parameter count. The QPA framework introduced in this work, by "generating" the parameters of PEFT methods, can be seamlessly integrated with other compression techniques, such as quantization and pruning. This combination enhances its practical applicability and opens promising avenues for further exploration of the QPA approach in the context of model compression.

## A.1    EFFICIENT LLM SELECTION FOR QPA

We provide a detailed rationale for selecting Gemma-2 and GPT-2 as the focus models. Prioritizing parameter-efficient models, we chose those that are both open-source and commercially accessible to ensure broad applicability and seamless integration into various systems. Given that QPA eliminates the need for quantum hardware during the inference stage, making QML more practical, we also focused on selecting classical models that are widely available and have manageable hardware requirements.

While the main text presents results for GPT-2 (80M), Table. 3 also includes QPA LoRA results for GPT-2 XL (1.5B), which exhibit similar trends, demonstrating QPA's ability to reduce training parameters while maintaining performance comparable to other PEFT methods.

Although the proposed QPA is applicable to any LLM and PEFT method (as outlined in Sec. 3), Gemma-2 (2B) was selected for this study as it offers the best balance between performance and resource efficiency given current computational constraints, whereas models with over 70B parameters remain beyond the scope of this work. Future research will explore the application of QPA to these larger models.

Table 3: Training parameters and testing perplexity for GPT-2 XL (1.5B) using LoRA and QPA methods, with emphasis on configurations achieving the most significant parameter reductions.

| PEFT Method | GPT-2 XL | |
| --- | --- | --- |
| | # Params (%) | PPL |
| LoRA | 0.12 | 1.474 |
| QPA LoRA setting 1 (Ours) | **0.07** | **1.485** |
| QPA LoRA setting 2 (Ours) | **0.09** | **1.475** |
| QPA LoRA setting 3 (Ours) | **0.13** | **1.469** |

## A.2    QUBIT NUMBER LIMITATIONS OF NON QUANTUM PARAMETER GENERATION METHODS

While it is possible to generate parameters for tuning tasks using conventional QML methods, that replacing the PEFT methods with VQC modules, a key difference lies in the output size. For an output of size $n$, conventional QML utilizes the expectation value of the Pauli-z operator, $\langle\psi(\boldsymbol{\theta})|\sigma_z^{(i)}|\psi(\boldsymbol{\theta})\rangle$, for each *qubit*, resulting in a qubit requirement of $n$, as described in Eq. 14 of (Mari et al., 2020). In contrast, quantum parameter generation leverages the measurement probability $|\langle\phi_i|\psi(\boldsymbol{\theta})\rangle|^2$ for each *basis*, reducing the qubit requirement to $\lceil\log_2 n\rceil$.

In practical terms, the qubit usage in the PEFT experiments presented in this study ranges between 4 and 11 qubits. If conventional QML were employed, the required qubit count would scale exponentially, from $2^4$ to $2^{11}$, an impractically large number that would make conventional QML infeasible for PEFT applications.

## B    Discussion on Error Performance

Prior works have examined the theoretical error performance of VQC in regression tasks (Qi et al., 2023) and classification tasks (Du et al., 2021b). For regression, the representational capacity is bounded by (Eq. 11 in (Qi et al., 2023)):

$$\mathcal{L}_{\mathcal{D}}(f_{\mathcal{D}}^*) = \|h_{\mathcal{D}}^*(\mathbf{x}) - \mathcal{T}_{lr}\left(\mathbb{E}\left[g(\mathbf{x}; \boldsymbol{\theta}_{vqc}, \boldsymbol{\theta}_{ttn})\right]\right)\|_1 \leq \frac{\Theta(1)}{\sqrt{U}} + \mathcal{O}(\frac{1}{\sqrt{M}}), \tag{11}$$

where $U$ represents the number of qubits and $M$ the number of measurement shots. While our quantum parameter generation framework diverges from traditional quantum machine learning methodologies, a promising direction for analyzing the approximation error of the QPA in estimating PEFT parameters may be guided by insights from the universal approximation theorem. Specifically, the representation power of pre-trained large language models (LLMs) could leverage population risk estimation, as outlined in previous theoretical work on parameter-efficient learning with Wasserstein measurements in Hilbert space (Yang et al., 2021b). In particular, the dimensionality of classical hidden neurons may align with that of the Hilbert space, utilizing fewer qubits and quantum gate parameters. This alignment suggests that a quantum approach may achieve comparable approximation accuracy with fewer parameters than classical counterparts as one perspective for future studies.

## C    Training Hyperparameter Configuration

In this section, we provide the training hyperparameter configuration used for the results presented in the main text. Notably, $\alpha$ represents the scaling factor in the low-rank adaptation methods. All experiments were conducted on NVIDIA V100S and NVIDIA H100 GPUs.

Table 4: Hyperparameter configurations of LoRA and QPA LoRA for fine-tuning GPT-2 and Gemma-2 with WikiText-2 dataset.

| Hyperparameters | LoRA | | QPA LoRA | |
|---|---|---|---|---|
| | GPT-2 | Gemma-2 | GPT-2 | Gemma-2 |
| $\alpha$ | 2r | | 2r | |
| Dropout | 0.05 | 0.0 | 0.05 | 0.0 |
| Optimizer | AdamW | | AdamW | |
| LR | 1e-5 | | 1e-5 | |
| LR Scheduler | Linear | | Linear | |
| Batch size | 1 | | 1 | |
| Warmup Steps | 0 | | 0 | |
| Epochs | 3 | 5 | 3 | 5 |

Table 5: Hyperparameter configurations of DoRA and QPA DoRA for fine-tuning GPT-2 and Gemma-2 with WikiText-2 dataset.

| Hyperparameters | DoRA | | QPA DoRA | |
|---|---|---|---|---|
| | GPT-2 | Gemma-2 | GPT-2 | Gemma-2 |
| $\alpha$ | 2r | | 2r | |
| Dropout | 0.0 | | 0.0 | |
| Optimizer | AdamW | | AdamW | |
| LR | 2e-6 | | 2e-6 | |
| LR Scheduler | Linear | | Linear | |
| Batch size | 1 | | 1 | |
| Warmup Steps | 100 | | 100 | |
| Epochs | 5 | | 5 | |

Table 6: Hyperparameter configurations of PT and QPA PT for fine-tuning GPT-2 and Gemma-2 with WikiText-2 dataset.

| Hyperparameters | PT | | QPA PT | |
|---|---|---|---|---|
| | GPT-2 | Gemma-2 | GPT-2 | Gemma-2 |
| Dropout | | 0.0 | | 0.0 |
| Optimizer | | AdamW | | AdamW |
| LR | | 5e-6 | | 1e-6 |
| LR Scheduler | | Linear | | Linear |
| Batch size | | 1 | | 1 |
| Warmup Steps | | 0 | | 0 |
| Epochs | | 5 | | 5 |

Table 7: Hyperparameter configurations of FFA and QPA FFA for fine-tuning GPT-2 and Gemma-2 with WikiText-2 dataset.

| Hyperparameters | FFA | | QPA FFA | |
|---|---|---|---|---|
| | GPT-2 | Gemma-2 | GPT-2 | Gemma-2 |
| Dropout | | 0.0 | | 0.0 |
| Optimizer | | AdamW | | AdamW |
| LR | | 5e-6 | | 5e-6 |
| LR Scheduler | | Linear | | Linear |
| Batch size | | 1 | | 1 |
| Warmup Steps | | 0 | | 0 |
| Epochs | 5 | 10 | 5 | 10 |

## D  QPA APPLIED AT DIFFERENT LoRA RANKS

In Sec. 4.1, to maintain clarity in the main text, we presented results only for QPA applied at LoRA and DoRA ranks of 4. In this appendix, we provide the full set of results in Fig. 5, showing QPA applied across various LoRA and DoRA ranks. Additionally, it can be observed that different values of $n_{mlp}$ may yield distinct optimal LoRA and DoRA ranks. Notably, the optimal LoRA rank identified in Sec. 4.2 was based on a fixed $n_{mlp}$ for simplicity.

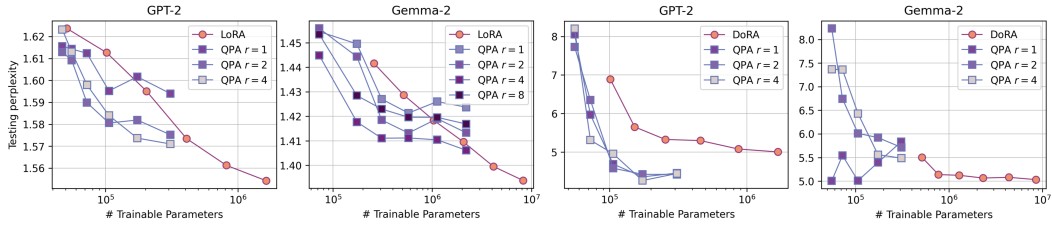

Figure 5: Full perplexity comparison of GPT-2 and Gemma-2 models versus the number of trainable parameters for LoRA, DoRA and QPA, evaluated on the WikiText-2 dataset. This figure shows the complete results with QPA applied across different LoRA and DoRA ranks, complementing the partial results presented in the main text.

## E  ON PENN TREEBANK DATASET

While the main text presents results obtained on the WikiText-2 dataset, we extend our analysis to showcase the broader applicability of QPA across additional datasets. In this section, we report results of QPA applied to LoRA, Prefix Tuning (PT), and Feed-Forward Adapter (FFA) on the Penn Treebank dataset. The findings indicate that QPA achieves its strongest advantage in the lower-

parameter regime for LoRA, consistently outperforms PT across all parameter ranges, and demonstrates comparable performance to FFA with a reduced number of trainable parameters. These observations align with the results reported in the main text for the WikiText-2 dataset.

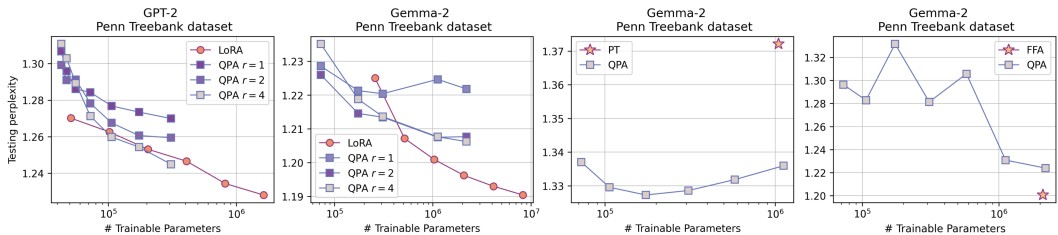

Figure 6: Testing perplexity of GPT-2 and Gemma-2 models versus the number of trainable parameters on the Penn Treebank dataset. The comparison includes LoRA and QPA applied at different LoRA ranks ($r = 1, 2$, and 4). The third subplot compares QPA with PT, and the fourth compares QPA with FFA, further illustrating QPA's competitive performance in various parameter configurations.

## F  EFFECTS OF DIFFERENT CIRCUIT ANSATZ

In the main context, the usage of the $R_Y$ circuit is inspired by the fact that the rotation $R_Y(\theta) = \begin{pmatrix} \cos(\frac{\theta}{2}) & -\sin(\frac{\theta}{2}) \\ \sin(\frac{\theta}{2}) & \cos(\frac{\theta}{2}) \end{pmatrix}$ produces quantum states with real amplitudes. This contrasts with $R_X(\theta) = \begin{pmatrix} \cos(\frac{\theta}{2}) & -i\sin(\frac{\theta}{2}) \\ -i\sin(\frac{\theta}{2}) & \cos(\frac{\theta}{2}) \end{pmatrix}$ and $R_Z(\theta) = \begin{pmatrix} e^{-i\frac{\theta}{2}} & 0 \\ 0 & e^{i\frac{\theta}{2}} \end{pmatrix}$, which involve complex numbers. Since the ultimate goal is to generate the parameters for the PEFT methods, which are typically real numbers, this rationale led us to select the $R_Y$ circuit in the main content.

While the quantum circuit ansatz in the main content is constructed as a combination of $R_Y$ and CNOT gates, it is also possible to explore alternative circuit ansatz to potentially improve performance. Similar to the construction of the $R_Y +$ CNOT ansatz in the main content (Eq. 1), alternative ansatz can be developed by replacing the parameterized $R_Y$ gate with other parameterized gates, such as $R_X$ :

$$|\psi(\boldsymbol{\theta})\rangle = \left( \prod_{i=1}^{N-1} \text{CNOT}^{i,i+1} \prod_{j=1}^{N} R_X^j(\theta_j^{(L)}) \right)^L |0\rangle^{\otimes N} \quad (R_X + \text{CNOT circuit}). \tag{12}$$

And the circuits combined with $R_Z$ gates:

$$|\psi(\boldsymbol{\theta})\rangle = \left( \prod_{i=1}^{N-1} \text{CNOT}^{i,i+1} \prod_{j=1}^{N} R_Z^j(\theta_j^{(L,Z)}) \prod_{i=1}^{N-1} \text{CNOT}^{i,i+1} \prod_{j=1}^{N} R_X^j(\theta_j^{(L,X)}) \right)^L |0\rangle^{\otimes N}$$

$$(R_X R_Z + \text{CNOT circuit}), \tag{13}$$

$$|\psi(\boldsymbol{\theta})\rangle = \left( \prod_{i=1}^{N-1} \text{CNOT}^{i,i+1} \prod_{j=1}^{N} R_Z^j(\theta_j^{(L,Z)}) \prod_{i=1}^{N-1} \text{CNOT}^{i,i+1} \prod_{j=1}^{N} R_Y^j(\theta_j^{(L,Y)}) \right)^L |0\rangle^{\otimes N}$$

$$(R_Y R_Z + \text{CNOT circuit}). \tag{14}$$

In Fig. 7, the LoRA results from Fig. 2 are extended to include an investigation of different circuit constructions, as described earlier. Notably, the overall performance differences are minimal, suggesting that the choice of circuit construction has only a minor impact under ideal conditions. The influence of more realistic conditions will be examined in detail in the following appendix section.

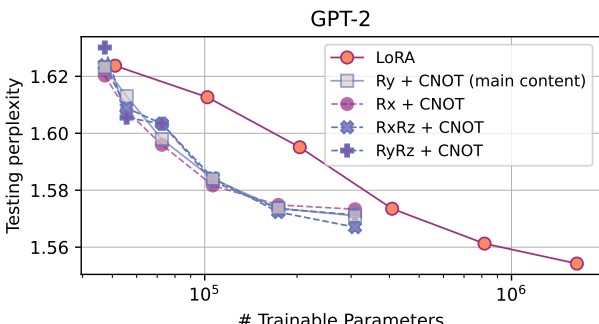

Figure 7: Testing perplexity of GPT-2 versus the number of trainable parameters on different circuit ansatz. The comparison includes LoRA and QPA applied at LoRA rank ($r = 4$).

## G   EFFECTS OF FINITE MEASUREMENT SHOTS AND NOISE

In the main content, the amplitudes and measurement probabilities of the basis states are computed exactly. Although the implementation on actual quantum hardware is left for future work, it is possible to simulate QPA performance on a real quantum computer by accounting for finite measurement shots and incorporating a noise model based on the real quantum hardware. On real quantum computers, measurement probabilities are derived from a finite number of measurement shots. The higher the number of shots, the more accurate the probability estimates. To understand the scaling behavior of the required measurement shots, we can draw parallels to tasks that approximate one quantum state with another. Based on theoretical findings from quantum fidelity tomography, the goal is to achieve a fidelity $F(|\gamma\rangle, |\varphi\rangle) \geq 1 - \epsilon$ between the output state $|\varphi\rangle$ and an unknown state $|\gamma\rangle$, with a given infidelity $\epsilon$. Prior work suggests that the sufficient number of measurement shots is

$$n_{\text{shot}} = O\left(\frac{2^N}{\epsilon} \log\left(\frac{2^N}{\epsilon}\right)\right) \tag{15}$$

(Haah et al., 2017), where $N$ represents the number of qubits.

As the number of measurement shots scales exponentially with the system size N, adopting a linear scaling strategy with respect to $2^N$ emerges as a practical approach to improve both training and testing performance in larger systems. In Fig. 8, the LoRA and QPA results for GPT-2 from Fig. 2 are analyzed under finite measurement shots. By comparing measurement shot counts $n_{\text{shot}} \in \{10 \times 2^N, 20 \times 2^N, 40 \times 2^N\}$, where $N$ is the number of qubits, it can be observed that increasing the number of measurement shots leads to improved results. As expected, the outcomes progressively approach the exact measurement results as the shot count increases. Specifically, the results for $n_{\text{shot}} = 40 \times 2^N$ are notably close to the exact measurement results in certain cases. The number of qubits $N$ in Fig. 8, corresponding to the progression from the smallest number of trainable parameters to the largest, is $(10, 9, 8, 7, 6, 5)$, as indicated in Fig. 4(a).

Consequently, the ability to swiftly gather measurement data is critical for the effective application of the QPA method. Generative models present a viable alternative by simulating measurement outcomes efficiently (Ahmed et al., 2021), which can alleviate the burden of needing exponentially large numbers of shots. By utilizing generative models, it becomes possible to reduce computational complexity and enhance the practicality of quantum state tomography in large-scale systems, thereby increasing the feasibility of applying QPA in real-world scenarios.

As discussed earlier, by simulating finite measurement shots and incorporating a noise model based on real quantum hardware, it is possible to analyze the performance behavior of QPA on an actual quantum computer. In this appendix section, we use noise models derived from the IBM quantum computers ibm_torino and ibm_fez in our finite measurement shot simulations [3].

---

[3]IBM Quantum provides noise models based on the properties of real hardware backends:https://docs.quantum.ibm.com/api/qiskit/0.19/qiskit.providers.aer.noise.NoiseModel.

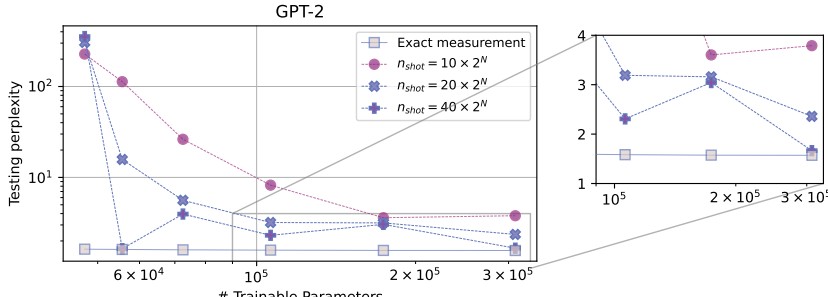

Figure 8: Testing perplexity of GPT-2 versus the number of trainable parameters on different number of measurement shots with $R_Y$ + CNOT ansatz. The comparison includes LoRA and QPA applied at LoRA rank ($r = 4$).

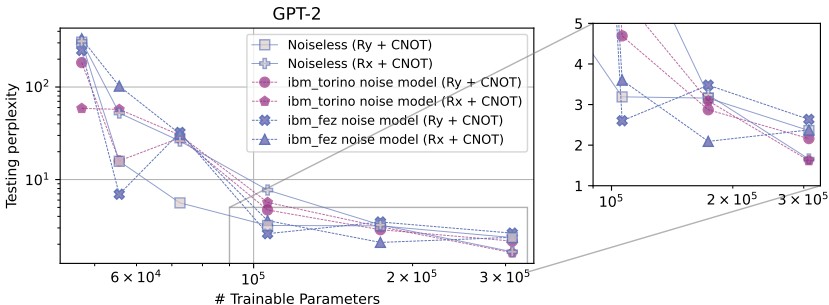

Figure 9: Testing perplexity of GPT-2 versus the number of trainable parameters on different noise setting with $R_Y$ + CNOT and $R_X$ + CNOT ansatz. The comparison includes LoRA and QPA applied at LoRA rank ($r = 4$), where $n_{\text{shot}}$ is fixed at $n_{\text{shot}} = 20 \times 2^N$.

In Fig. 9, the QPA performance results are presented in the presence of quantum computer noise models for two circuit constructions: $R_X$ + CNOT and $R_Y$ + CNOT. In the noiseless case, the $R_Y$ circuit outperforms the $R_X$ circuit in overall trends. Interestingly, when noise from ibm_torino and ibm_fez is introduced, although performance decreases slightly in some cases, it improves in most instances. This phenomenon aligns with previous observations in some quantum computing studies, where quantum noise has been shown to enhance performance in certain paradigms (Domingo et al., 2023).

Similarly, in classical machine learning, studies have demonstrated that a small amount of noise can be beneficial for fine-tuning LLMs (Wu et al., 2022). Since the quantum circuit in QPA generates measurement probabilities that are subsequently input into a classical MLP mapping model to produce PEFT parameters, this suggests that quantum noise may play a role analogous to the "small noise" effect observed in prior studies, aiding in the fine-tuning of LLMs, exploring more parameter spaces.

## H    ON GRADIENT VARIANCE OF QUANTUM CIRCUIT PARAMETERS

The occurrence of barren plateaus is a critical challenge in the training of QNNs (McClean et al., 2018; Zhang et al., 2022). Barren plateaus typically arise in learning tasks where the objective is the expectation value of some Hermitian operator $H$, expressed as:

$$E(\theta) = \langle 0|U(\theta)^{\dagger}HU(\theta)|0\rangle. \tag{16}$$

In contrast, our QPA approach deviates from this framework. The objective of QPA is the loss function of a target classical model (LLM, in this study). The parameters of this target model are updated using PEFT methods, with these parameters being generated by a mapping model that processes information derived from the measurement results of the QNN. Consequently, the output

of our QNN is the measurement result of basis states, rather than the expectation value of a Hermitian operator. As a result, whether the exponential vanishing gradient characteristic of barren plateaus occurs in QPA remains an open question.

To investigate whether barren plateaus are present in QPA, we analyzed the variance of the QNN gradient $\partial\theta$ in the QPA-LoRA results for GPT-2, as shown in Fig. 10. Notably, no significant downward trend in gradient variance is observed with increasing qubit count. This behavior may stem from the fact that QNN gradients are propagated backward through the subsequent mapping model, preventing them from behaving like the expectation value-based objectives typically seen in traditional QML.

Interestingly, however, as the QNN repetition $L$ increases, a slight downward trend in gradient variance is observed. This behavior bears some resemblance to both classical deep feedforward neural networks and barren plateaus in QNNs, as described in (Glorot & Bengio, 2010; McClean et al., 2018), albeit to a much lesser extent. This suggests that, while QPA does not exhibit barren plateaus in the same way as traditional QML (specifically with respect to qubit counts), its gradient dynamics may still reflect certain characteristics of both quantum and classical learning paradigms.

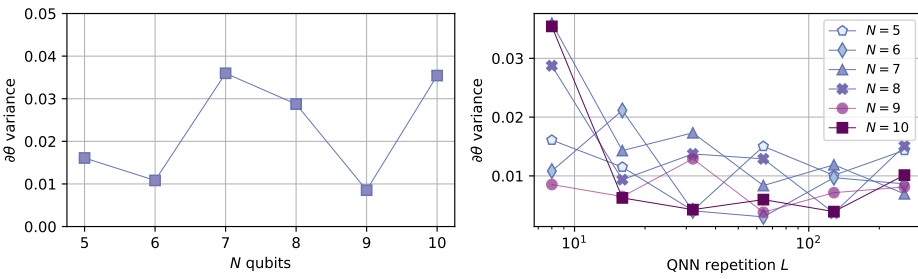

Figure 10: Variance of the gradient $\partial\theta$ with respect to the number of qubits $(N)$ and the QNN repetition depth $(L)$.

## I    ON COMPUTATIONAL TIME

The total execution times for LoRA and QPA, as presented in Fig. 2, are detailed in Table 8 and Table 9. For the LoRA results in Fig. 2, the average batch execution times for classical LoRA and QPA are 0.0154s and 0.0460s for GPT-2, and 0.0207s and 0.0576s for Gemma-2, respectively. Therefore, when accounting for the additional quantum circuit simulation and computational overhead of QPA, its average execution time is approximately three times that of LoRA.

Based on this observation, we can compare the performance of LoRA and QPA under similar total computational time. Specifically, for every epoch of QPA training, LoRA can complete approximately 3 epochs. In Fig. 11, we compare the performance of QPA and LoRA under different numbers of training epochs. For GPT-2, since both LoRA and QPA originally undergo 3 epochs of training, the QPA results with 1 epoch serve as a benchmark for comparing QPA's performance under a similar total execution time to that of LoRA.

For Gemma-2, the analysis follows a similar rationale. With 5 training epochs set as the baseline, we provide results for 1 and 2 epochs of QPA to estimate its performance under a total execution time comparable to LoRA. The results show that, although QPA with a similar total execution time does not outperform LoRA to the same extent as when both methods have the same number of epochs, it still exhibits better performance in regions with smaller parameter sizes.

## J    IMPLEMENTATION DETAIL

This section provides a minimal example of how QPA is applied to LoRA in a Gemma-2 model, implemented using PyTorch and TorchQuantum. The code snippets demonstrate the key components

Table 8: Training parameters and total execution time for GPT-2 using LoRA and QPA methods. Corresponding to the results provided in Fig. 2

| | GPT-2 LoRA | | GPT-2 QPA | |
|---|---|---|---|---|
| # Params | Total execution time (s) | # Params | Total execution time (s) | |
| 51025 | 1696 | 47248 | 5965 | |
| 102050 | 1677 | 55656 | 5520 | |
| 204100 | 1695 | 72512 | 5223 | |
| 408200 | 1712 | 106264 | 4910 | |
| 816400 | 1700 | 173808 | 4570 | |
| 1632800 | 1729 | 308936 | 4245 | |

Table 9: Training parameters and total execution time for Gemma-2 using LoRA and QPA methods. Corresponding to the results provided in Fig. 2

| | Gemma-2 LoRA | | Gemma-2 QPA | |
|---|---|---|---|---|
| # Params | Total execution time (s) | # Params | Total execution time (s) | |
| 258048 | 3796 | 72592 | 13861 | |
| 516096 | 3794 | 173888 | 11602 | |
| 1032192 | 3798 | 309016 | 8316 | |
| 2064384 | 3690 | 575154 | 9252 | |
| 4128768 | 4002 | 1119944 | 11768 | |
| 8257536 | 3816 | 2201248 | 8702 | |

of the QPA process, including the modification of the target layer and the generation of low-rank matrices using a quantum parameter generation approach.

```python
class QPA_LoRAGemma2(nn.Module):

    def __init__(self, gemma_model):
        super(QPA_LoRAGemma2, self).__init__()

        self.gemma_model = gemma_model

        for name, module in self.gemma_model.named_modules():
            if name == 'lm_head':
                lora_layer = QPA_LoRALayer(module, r=LoRA_rank, alpha=LoRA_rank*2)
                setattr(self.gemma_model, name, lora_layer)

    def forward(self, input_ids, attention_mask, labels):

        outputs = self.gemma_model(input_ids, attention_mask=attention_mask, labels =
          labels)

        return outputs
```

In the above example, the "lm_head" layer is modified using the QPA_LoRALayer function, defined as follows:

```python
class QPA_LoRALayer(nn.Module):
    def __init__(self, original_layer, r, alpha):
        super(QPA_LoRALayer, self).__init__()
        self.original_layer = original_layer
        self.r = r
        self.alpha = alpha
        self.dropout = nn.Dropout(p=0.0)
        self.dtype = torch.float32  # Ensure all tensors are of this type

        # Generate the parameters of A and B
        self.QPA_res = nn.ModuleList([
            QPA_Net(
                original_layer.weight.size(0)*r + r*original_layer.weight.size(1),
                1)
            ]).cuda()
```

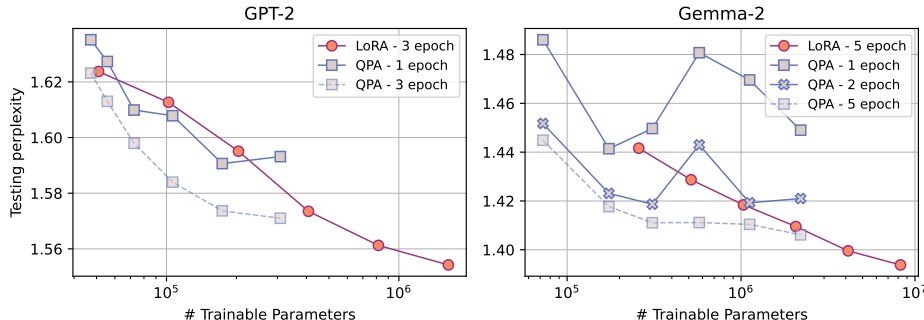

Figure 11: Comparison of testing perplexity against the number of trainable parameters for GPT-2 (left) and Gemma-2 (right) models using LoRA and QPA approaches across different epochs.

```
16
17
18              # Freeze the original layer's parameters
19              for param in self.original_layer.parameters():
20                  param.requires_grad = False
21
22          def forward(self, x):
23
24              gen_weights = []
25              for sub_res in self.QPA_res:
26                  gen_weights.append(sub_res())
27              self.generated_weights = torch.cat(gen_weights,
                ↪   dim=0).view(-1)[:self.original_layer.weight.size(0)*self.r +
                ↪   self.r*self.original_layer.weight.size(1)].cuda()
28              self.generated_weights_A =
                ↪   self.generated_weights[:self.original_layer.weight.size(0)*self.r]
29              self.generated_weights_A =
                ↪   self.generated_weights_A.view(self.original_layer.weight.size(0),
                ↪   self.r).type(self.dtype)
30              self.generated_weights_B =
                ↪   self.generated_weights[self.original_layer.weight.size(0)*self.r:]
31              self.generated_weights_B = self.generated_weights_B.view(self.r,
                ↪   self.original_layer.weight.size(1)).type(self.dtype)
32
33              batch_size, seq_len, hidden_size = x.size()
34              x_reshaped = self.dropout(x).view(-1, hidden_size)
35              delta = (x_reshaped @ self.generated_weights_B.t()) @ self.generated_weights_A.t()
36              delta = delta * (self.alpha / self.r)
37              delta = delta.view(batch_size, seq_len, self.generated_weights_A.shape[0])
38              return self.original_layer(x) + delta
39
```

As shown in lines 28 and 30, the low-rank matrices $A$ and $B$ are generated by the QPA_Net object, which implements the method described in Sec. 3. The QPA_Net class is defined as follows:

```
1   class QPA_Net(nn.Module):
2       def __init__(self, vocab_size, hidden_size):
3           super(QPA_Net, self).__init__()
4
5           self.n_sub_res = 1
6           self.weight_length = int(np.ceil((vocab_size * hidden_size) / self.n_sub_res ))
7
8           self.out_dim_mlp = 32
9           self.out_dim_MLP = chunk_size
10          self.batch_size = int(np.ceil((self.weight_length/self.out_dim_MLP)))
11          self.dropout = nn.Dropout(p=0.)
12
13          self.device = torch.device("cuda" if torch.cuda.is_available() else "cpu")
14
15          self.init_mapping = "MLP"
16          self.classical_layers = "MLP"
17
18
19          self.n_qubit_qpa = int(np.ceil(np.log2(self.batch_size)))
```

```
20          self.n_qubit = self.n_qubit_qpa
21          self.q_depth   = qnn_depth
22          self.QuantumNN = QLayer(self.q_depth, self.n_qubit_qpa).to(self.device)
23
24          if self.init_mapping == "MLP":
25              self.MappingNetwork = MappingModel(self.n_qubit+1, [32, 64, 128, 128, 64, 32],
                ↪   self.out_dim_mlp)
26
27
28          if self.classical_layers == "MLP":
29              self.fc1 = nn.Linear(self.out_dim_mlp, self.out_dim_MLP)
30
31
32      def forward(self):
33
34          compute_method = "checkpoint"
35
36
37          probs_ = self.QuantumNN().flatten()
38          probs_ = probs_[:self.batch_size]
39          probs_ = probs_.reshape(self.batch_size, 1, 1)
40
41
42          qubit_states_torch = generate_qubit_states_torch(self.n_qubit,
            ↪   self.batch_size)[:self.weight_length].to(self.device)
43
44          combined_data_torch = torch.cat((qubit_states_torch, probs_), dim=2)
45
46          prob_val_post_processed_list = []
47          if compute_method == "checkpoint":
48
49
50              batch_data = combined_data_torch[0:self.batch_size]
51              batch_data.requires_grad_()
52
53              prob_val_post_processed_batch = checkpoint(self.MappingNetwork, batch_data)
54
55              if self.classical_layers == "MLP":
56
57                  prob_val_post_processed_batch = checkpoint(self.dropout,
                    ↪   prob_val_post_processed_batch)
58                  prob_val_post_processed_batch = checkpoint(self.fc1,
                    ↪   prob_val_post_processed_batch)
59
60
61              prob_val_post_processed_list.append(prob_val_post_processed_batch)
62
63              torch.cuda.empty_cache()
64
65          prob_val_post_processed_list = prob_val_post_processed_list[:self.weight_length]
66          prob_val_post_processed = torch.cat(prob_val_post_processed_list, dim=0)
67
68          prob_val_post_processed = prob_val_post_processed.view(-1)[:self.weight_length]
69          prob_val_post_processed = prob_val_post_processed - prob_val_post_processed.mean()
70
71          torch.cuda.empty_cache()
72
73          return prob_val_post_processed
```

This implementation demonstrates how QPA is applied to generate low-rank matrices using quantum parameter genreation, which can be seamlessly integrated into LLMs to improve parameter efficiency.

