# OpenReview forum: "A Quantum Circuit-Based Compression Perspective for Parameter-Efficient Learning"
_ICLR.cc/2025/Conference — ICLR 2025 Poster_

### Official Review · Reviewer_KJS2 · 2024-11-02

**Soundness:** 2
**Presentation:** 3
**Contribution:** 2
**Rating:** 6
**Confidence:** 4

**Summary:**

The paper presents a novel approach for quantum parameter generation aimed at providing a quantum-classical solution for fine-tuning LLM's parameters.  Authors conduct simulation on benchmark datasets to assess performance compared to traditional fine-tuning techniques like LoRA and DoRA. Results indicate improvements in reducing the number of parameters while maintaining or surpassing the performance of existing PEFT methods.

**Strengths:**

The authors introduce an new framework that aims to bridge the gap between classical machine learning and quantum computing.

**Weaknesses:**

1. Efficiency Concerns: The proposed quantum parameter generation method, while designed to reduce the need for quantum hardware during inference, still relies heavily on quantum neural networks (QNN) during training. This raises critical questions regarding the efficiency of the overall process. Specifically, the optimization of quantum neural network parameters is prone to challenges such as barren plateaus, which can significantly hinder the training process. The authors do not sufficiently address how these issues may impact the efficiency of fine-tuning parameters of classical large models, particularly given the complexities associated with high-dimensional optimization landscapes typical of QNN.

2. The inputs of the mapping model necessitates a precise value of the corresponding measurement probabilities of the basis of quantum states. Typically, these probabilities can only be approximated through repeated measurements and averaging, which introduces another layer of complexity. The need for multiple measurements raises concerns regarding the time overhead involved. The paper does not adequately discuss the potential delays that may arise from extensive measurement requirements, especially in the context of noisy intermediate-scale quantum (NISQ) devices where measurement fidelity can be compromised by inherent noise. Furthermore, the implications of limited measurement shots on the accuracy of estimated probabilities and the subsequent effect on LLM's fine-tuning are not sufficiently explored.

Although the authors mention the limitations of finite measurement shots in Appendix B, the reviewer believes that providing sufficient experimental results, including finite measurements and noise-affected simulations, is essential for demonstrating the feasibility of the proposed method in the paper.

**Questions:**

Reviewer would be grateful if the authors could discuss the following:

Given that the primary idea of this paper aims to leverage the exponential dimensionality of quantum states to represent the parameter space of LLMs. There exists a parallel in tensor network [1,2] theory that serves a similar purpose: the most successful application of tensor networks in physics has been to approximate exponentially large vectors arising in quantum mechanics.

The question is that might it also be feasible to explore the application of tensor network's theories and methodologies as an alternative to the currently resource-constrained and large-scale unavailable quantum computing?

Additionally, could the authors utilize certain theoretical insights (e.g. from tensor network research) to investigate why the proposed method achieves superior performance compared to traditional fine-tuning techniques such as LoRA and DoRA, rather than relying solely on empirical evidence?

This discussion could significantly enhance the theoretical underpinnings of the paper and provide a more robust framework for understanding the advantages of the proposed approach.

[1] Stoudenmire E, Schwab D J. Supervised learning with tensor networks[J]. Advances in neural information processing systems, 2016, 29.

[2] Liu Z, Yu L W, Duan L M, et al. Presence and absence of barren plateaus in tensor-network based machine learning[J]. Physical Review Letters, 2022, 129(27): 270501.

---

> ### Author Response · Authors · 2024-11-21
>
> We sincerely thank Reviewer KJS2 for the invaluable feedback and suggestions, which have significantly enhanced the quality of our work. We greatly appreciate the time and effort dedicated to providing such thoughtful insights, which have been crucial in helping us refine and improve our paper.
>
> >**W1**. Efficiency Concerns: ... landscapes typical of QNN.
>
> **Response**: We thank the reviewer for their thoughtful comment. In response to the concerns regarding challenges like barren plateaus, which can significantly hinder the training process, we have added a new appendix H section titled “On Gradient Variance of Quantum Circuit Parameters” to address this issue.
>
> In the original study [1], barren plateaus are associated with loss functions defined as the expectation value of Hermitian operators. In contrast, QPA evaluates the loss of a target classical model (an LLM in this study), with updates generated using PEFT methods. The QNN output provides measurement probabilities rather than operator expectations, which distinguishes our approach.
>
> Our analysis of QNN gradient variance, $\partial \theta$, in QPA-LoRA results for GPT-2 (Fig. 10) reveals no significant downward trend as the number of qubits increases. This stability is likely due to the backward propagation of gradients through the mapping model, mitigating the exponential vanishing trends typically seen in barren plateaus. However, we observed a slight downward trend with increasing QNN repetitions $L$, resembling patterns seen in deep feedforward networks and barren plateaus [1,2], though to a much lesser extent. These findings suggest that QPA’s gradient dynamics share characteristics of both quantum and classical paradigms without exhibiting severe barren plateau effects.
>
> > **W2**. The inputs of ... sufficiently explored.
>
> **Response**: To explore the impact of limited measurement shots on the accuracy of estimated probabilities and their subsequent effect on fine-tuning LLMs, we have added Appendix G, “Effects of Finite Measurement Shots and Noise,” to the revised version. As illustrated in the newly added Fig. 8, increasing the number of measurement shots improves performance, eventually converging toward exact simulation results. While practical implementations must carefully balance measurement budgets and performance, our method demonstrates effectiveness even with a finite number of shots, highlighting its utility beyond idealized strong simulations.
>
> Additionally, in this appendix, we incorporate noise models provided by IBM, based on real hardware data from ibm_torino and ibm_fez, to investigate the potential behavior of QPA in real-device environments. Interestingly, introducing noise from ibm_torino and ibm_fez yields mixed results: while performance decreases slightly in some cases, it improves in most instances. This observation is consistent with prior quantum computing studies, where quantum noise has been shown to enhance performance in specific paradigms [3].
>
> In a similar way, classical machine learning studies have found that small amounts of noise can improve LLM fine-tuning [4]. Since the quantum circuit in QPA generates measurement probabilities that are input into a classical MLP mapping model to produce PEFT parameters, it is plausible that quantum noise plays a similar role to the “small noise” effect observed in these studies. Such noise may facilitate exploration of a broader parameter space, ultimately aiding in the fine-tuning of LLMs.
>
> > **Q1**. The question is that ... unavailable quantum computing?
>
> **Response**: We thank the reviewer for raising this intriguing question. From the perspective of quantum-inspired methods, fast quantum simulation or approximation techniques, such as tensor networks, offer a feasible way to represent the state of a quantum circuit. However, these methods typically require circuits with limited entanglement or shallow depths to ensure that the quantum state remains sufficiently simple for approximation.
>
> In contrast, the quantum circuit proposed in this study is designed for greater generality, enabling the construction of complex and deep quantum neural networks. At the current stage, we believe that tensor networks may be a viable alternative in scenarios where the dimensionality of the tuning target is inherently low. For instance, if the “update” of the weight matrix during model training—the tuning target—is simple enough to be represented by low-dimensional structures, tensor networks could be a valuable tool to explore within the QPA framework. This remains an interesting avenue for future investigation.

---

> ### Author Response · Authors · 2024-11-21
>
> > **Q2**. Additionally, could the authors ... empirical evidence?
>
> **Response**: We thank the reviewer for this insightful question. While a comprehensive theoretical analysis of QPA is planned for future work, this study primarily focuses on demonstrating the feasibility and potential of QPA for PEFT in large-scale machine learning tasks, particularly billion-parameter LLMs. In Appendix B, we outline possible directions for developing a theoretical framework for QPA, building on existing studies of VQC and QNN theories.
>
> We also appreciate the reviewer highlighting tensor networks as a potential link to the theoretical foundation of QPA. Tensor networks are highly effective in decomposing high-dimensional tensors into a network of smaller interconnected tensors, capturing local correlations while discarding irrelevant degrees of freedom. Given that quantum circuit states can also be approximated by tensor networks (with accuracy depending on the rank), the quantum circuit used in QPA can be viewed as a more general and flexible representation of high-dimensional tensors. This flexibility enables a richer representation of the parameters in PEFT methods.
>
>
>
> While the development of a full theoretical framework remains an important goal for future work, the empirical results presented in this study provide a solid foundation. These findings demonstrate the practical viability of QPA and its scalability to large models, serving as an essential step toward understanding and optimizing QPA for long-term effectiveness.
>
>
> [1] Jarrod R McClean, et al. Barren plateaus in quantum neural network training landscapes. Nature communications, 9(1):4812, 2018.
>
> [2] Xavier Glorot and Yoshua Bengio. Understanding the difficulty of training deep feedforward neural networks. In Proceedings of the thirteenth international conference on artificial intelligence and statistics, pp. 249–256. JMLR Workshop and Conference Proceedings, 2010.
>
> [3] Laia Domingo, G Carlo, and F Borondo. Taking advantage of noise in quantum reservoir computing. Scientific Reports, 13(1):8790, 2023.
>
> [4] Chuhan Wu, et. al. Noisytune: A little noise can help you finetune pretrained language models better. arXiv preprint arXiv:2202.12024, 2022.

---

> ### Author Response · Authors · 2024-11-25
> **A Kind Reminder**
>
> Dear Reviewer,
>
> With the rebuttal phase nearing its conclusion, we would like to gently remind you of our responses.
>
> We have thoughtfully addressed all your comments and questions, and we hope our revisions meet your expectations.
>
> If there are any unresolved concerns, we are more than willing to discuss them further. Otherwise, if you find our updates satisfactory, we kindly invite you to consider raising your score.
>
> Thank you again for your time and valuable feedback!

---

> > ### Comment · Reviewer_KJS2 · 2024-11-25
> >
> > Thank you for the author's feedback. I appreciate the detailed responses provided in your rebuttal, which have clarified most of my initial concerns. The explanations in Appendix F and G regarding the choice of quantum ansatz and the effects of noise have strengthened the paper, and the empirical comparison with LoRA in Appendix I provides valuable insights into the simulation computational trade-offs. These revisions have enhanced the clarity and depth of your submission. Based on this, I am happy to raise my score to 6 and increase my confidence.
> >
> > However, a key concern remains regarding the practical economic and computational efficiency of QPA. The reliance on quantum neural networks (QNNs) for fine-tuning large models, using methods like parameter shift for gradient decent on a real quantum computer, scales linearly with the number of parameters, leading to significant computational costs. This raises doubts about its feasibility on current quantum hardware, which faces resource limitations and noise challenges. Furthermore, the necessity of QNNs over advanced classical methods, such as optimized low-rank adaptations, requires clearer justification. Addressing these trade-offs against classical alternatives would enhance the paper’s practical relevance and impact.

---

> > > ### Author Response · Authors · 2024-11-25
> > >
> > > We are pleased that most of the reviewer’s concerns have been addressed, enhancing the clarity and depth of our submission. Regarding the practical economic and computational efficiency of QPA, we note that the reviewer’s concern reflects a broader challenge associated with NISQ hardware. As quantum hardware continues to advance, issues such as noise and measurement overhead are expected to diminish. However, the conceptual framework and potential of QPA for generating parameter-efficient representations of PEFT methods remain valid and promising as quantum technology evolves.
> > >
> > > We are also grateful for the insightful comment on addressing the trade-offs with classical alternatives, which provides a compelling direction for future work. Since QPA is designed to “generate” parameters, it offers the potential not only to compete with classical approaches but also to integrate efficiently with them. Exploring such hybrid combinations could be an intriguing area of future research.
> > >
> > > Once again, we would like to express our heartfelt appreciation for the reviewer’s time and effort in thoroughly reviewing our paper and providing invaluable feedback.

---

### Official Review · Reviewer_Y4nw · 2024-11-03

**Soundness:** 3
**Presentation:** 3
**Contribution:** 2
**Rating:** 6
**Confidence:** 4

**Summary:**

The paper introduces Quantum Parameter Adaptation (QPA), a novel approach for parameter-efficient learning that combines quantum neural networks (QNNs) with classical machine learning to generate parameters for fine-tuning large models. The QPA method utilizes quantum circuits to generate classical model weights, allowing for significant parameter
reduction without compromising performance, as demonstrated with Gemma-2 and GPT-2 case studies. The method enables practical use of quantum machine learning by decoupling quantum hardware requirements from the inference phase, thus making inference feasible on classical systems. The results show that QPA can effectively reduce the number of parameters required for methods like Low-Rank Adaptation (LoRA), while maintaining or improving model performance, indicating its potential as a scalable and efficient quantum-classical hybrid solution for large-scale learning tasks.

**Strengths:**

1. The paper proposes the Quantum Parameter Adaptation (QPA) method, which is a new method that combines  quantum neural networks (QNN) with classical multi-layer perceptrons to generate The parameters of the fine-tuning method demonstrate the efficiency in parameter reduction.
2. QPA separates quantum computing from the inference stage, allowing the model to perform inference without  relying on quantum hardware, greatly reducing deployment complexity and thus having practical application value for classical hardware inference.
3. Experimental results show that QPA can significantly reduce the number of parameters while maintaining or slightly improving model performance. In contrast, methods such as LoRA, DoRA,  and Prefix Tuning (PT) are highly efficient in parameters. The performance in learning tasks is not as good as QPA, highlighting its high efficiency.
4. The authors extend the QPA method to large models such as GPT-2 and Gemma-2, significantly expanding the scope of previous quantum parameter generation studies.
5. QPA allows flexible exploration of different parameter configurations, improving its
adaptability in different models and parameter efficient fine-tuning (PEFT) methods.

**Weaknesses:**

1. The study assumes that the measurement probability is exact and does not consider noise, which may not accurately reflect the situation where a finite number of measurements are required in the actual quantum hardware environment. Although the impact of this limitation on performance is discussed, the problem is not completely solved. It is possible to consider introducing some simulated noise models to verify the robustness of the proposed QPA.
2. The paper mentions that QPA uses RY and CNOT gates to build quantum circuit ansatz, but lacks a detailed explanation of why this ansatz structure is chosen. It is recommended that the author add a detailed argument for the choice of ansatz and compare the effects of different quantum gate combinations to ensure the rationality of the current choice.
3. The method uses the gradient obtained from quantum measurements, which may introduce additional complexity in the training process, especially when implemented on actual quantum hardware, due to the presence of parameter offsets and measurement noise.
4. The paper mentions that the theoretical analysis of the convergence behavior, trainability and learnability characteristics of QPA will be addressed in future research, which shows that there is a gap in the current understanding of the long-term effectiveness of QPA.
5. More shortcomings can be found in the Questions section.

**Questions:**

1. Lack of comparison with other rotation gates: The author mainly chose the combination of RY gate and CNOT gate, but there is a lack of detailed comparative analysis in the article. Why is the RY gate more suitable for this specific task, while the RX and RZ gates are not good enough? The author needs relevant experiments to verify the advantages of the RY gate over other gates.
Rationality of the selection of rotation gates: RX, RY and RZ gates are all single-bit rotation gates, why does RY have a better effect in this scenario? Is there a clear mathematical or experimental basis to show that the efficiency and effect of the RY gate in quantum parameter generation is better than RX or RZ? If not, does this mean that the author's choice of the RY gate is more based on experience rather than theoretical support?
2. Performance in noisy environments: Although the RY gate is considered to perform better in capturing the rotation characteristics of quantum states, quantum hardware has noise and errors in actual operation. Have you compared the robustness of different rotation gates in actual quantum hardware? Especially when considering quantum noise, is the RY gate still the best choice? At the same time, the performance of QPA in actual quantum hardware remains to be considered.
3. Lack of discussion on time complexity: Although QPA performs well in reducing the number of parameters, the paper lacks a detailed discussion on the running time of the quantum circuit when generating parameters. Does the generation and measurement process of the quantum circuit bring additional time overhead? Does this additional time complexity offset the improvement in training efficiency brought by parameter reduction?
Compared with the classical PEFT method, is there a significant time overhead in the process of QPA using quantum circuits to generate parameters? Do these differences have a significant impact on the practical application of models at different scales? In practical applications, the time efficiency of the model is one of the important considerations. Even if the number of parameters is reduced, if the time complexity of the quantum circuit is high when generating parameters, this may affect the efficiency and scalability of the overall system. The author should provide more quantitative analysis to evaluate the overall cost-effectiveness of the QPA method in terms of time and computing resources.

4. In more complex scenarios, how does QPA compare to traditional QML methods? : The paper shows the efficiency of QPA in reducing parameters of large models such as GPT-2 and Gemma-2. However, how does QPA perform when extended to larger models such as GPT-3 or larger? Does quantum parameter generation encounter scalability issues in such scenarios?

5. Best choice of parameter configuration: How to determine the best block size (nmlp) and number of QNN repetitions (L) for different PEFT methods? Is there a general method to determine these hyperparameters for other models?

---

> ### Author Response · Authors · 2024-11-21
>
> We are truly grateful to Reviewer Y4nw for the invaluable suggestions, which have greatly contributed to improving the quality of our work. We deeply appreciate the time and effort invested in providing thoughtful feedback, which has been instrumental in helping us refine and enhance our paper.
>
> > **W1**. The study assumes ... of the proposed QPA.
>
> **Response**: We thank the reviewer for the insightful comment.  Indeed, to consider noisy environments and finite measurement shots can better reflect the behavior of QPA method in a more realistic way. Thus, in the revised version, we have added Appendix G “Effects of Finite Measurement Shots and Noise” to address this topic.
>
> In this appendix section, we utilize the noise model provided by IBM, based on real hardware data from ibm_torino and ibm_fez  to investigate the potential behavior of QPA in real-device environments.
>
> Interestingly, introducing noise from ibm_torino and ibm_fez leads to mixed outcomes: while performance decreases slightly in some cases, it improves in most instances. This aligns with previous findings in quantum computing studies, where quantum noise has been shown to enhance performance in certain paradigms [1].
>
> Similarly, in classical machine learning, studies have demonstrated that small amounts of noise can improve the fine-tuning of LLMs [2]. Given that the quantum circuit in QPA generates measurement probabilities, which are then fed into a classical MLP mapping model to produce PEFT parameters, it is plausible that quantum noise serves a similar role to the “small noise” effect observed in prior studies. This noise may help explore a broader parameter space, ultimately aiding in the fine-tuning of LLMs.
>
> > **W2**. The paper mentions ... of the current choice.
>
> **Response**: We sincerely appreciate the reviewer’s thoughtful question, which has provided valuable insights and helped us improve the context of our work. The choice of  $R_Y$  and CNOT gates is motivated by the ultimate goal of generating parameters for PEFT methods, which are typically real numbers. This rationale underpins our decision to use the  $R_Y$  circuit in the main content.
>
> In response to the reviewer’s suggestion, we have added Appendix F, “Effects of Different Circuit Ansatz,” to the revised version to explore this topic further. Alongside the  $R_Y$  and CNOT combination, we examined  $R_X$ + CNOT ,  $R_Y R_Z$ + CNOT, and  $R_X R_Z$ + CNOT  configurations for generating LoRA parameters in the QPA task (Fig. 7). The results show no significant performance differences between these circuit constructions. However, when a finite number of measurement shots is considered in a noiseless environment (Fig. 9), the  $R_Y$ + CNOT  combination demonstrates superior performance compared to the  $R_X$ + CNOT  combination, suggesting that $R_Y$ + CNOT is indeed a reasonably effective choice.
>
> > **W3**. The method uses ... measurement noise.
>
> **Response**: We thank the reviewer for the insightful comment. It is true that gradient computation from quantum measurements can introduce additional complexity in the training process due to realistic hardware noise. As addressed in our response to w1, the revised version includes an investigation into the effect of noise on QPA using noise models derived from real quantum computers. Our findings indicate that QPA exhibits minimal sensitivity to noise and, in some cases, even demonstrates improved performance under noisy conditions.
>
> > **W4**. The paper mentions ... effectiveness of QPA.
>
> **Response**: We sincerely thank the reviewer for raising this important point. While it is true that the theoretical analysis of convergence behavior, trainability, and learnability characteristics of QPA is deferred to future work, the current study primarily focuses on demonstrating the feasibility and potential of QPA for PEFT in large-scale machine learning tasks, particularly for billion-parameter LLMs.
>
> To address this gap, we have included several empirical analyses to provide insights into the method’s effectiveness. For instance, in the newly added Appendix H, we investigate gradient variance and demonstrate that QPA does not exhibit the exponential vanishing gradient phenomenon, often referred to as barren plateaus, typically associated with QNNs as the number of qubits increases. Furthermore, the robustness of QPA to noise, as analyzed in Appendix G, underscores its potential applicability to real-world quantum systems, even under non-ideal conditions.
>
> While a comprehensive theoretical framework is indeed an important direction for future work, the empirical results provided in this study lay a strong foundation, demonstrating the practical viability of QPA and its ability to scale to large models. This serves as an essential step toward understanding and optimizing QPA for long-term effectiveness.

---

> ### Author Response · Authors · 2024-11-21
>
> > **Q1**. Lack of comparison ... rather than theoretical support?
>
> **Response**: We thank the reviewer for this question. As mentioned in our response to w2, we have added Appendix F, “Effects of Different Circuit Ansatz,” to the revised version to explore this topic further.
>
> > **Q2**. Performance in noisy ... be considered.
>
> **Response**: We thank the reviewer for this question. As noted in our response to w1, we have added Appendix G, “Effects of Finite Measurement Shots and Noise,” to the revised version to investigate this topic further. While these results are based on simulations rather than actual hardware, the noise model used is derived from real hardware data and should provide a reasonable approximation of QPA’s behavior on real devices.
>
> > **Q3**. Lack of discussion on ... and computing resources.
>
> **Response**: We appreciate the reviewer’s comment regarding computational time, as it highlights an important aspect of our work. In response, we have added Appendix I, “On Computational Time,” to the revised version to investigate this topic in detail.
>
> While reducing the number of training parameters in QPA does require additional computations for quantum operations and the mapping model, we anticipate that improvements in quantum hardware and operation speed will help mitigate this challenge in the future. To compare the computational cost of QPA with classical approaches at the current stage of simulation, we have included execution time data in the newly added Appendix I.
>
> From the LoRA results in Fig. 2, we observe that the average execution time of QPA is approximately three times longer than that of classical LoRA. This means that while QPA completes one epoch, LoRA can complete three epochs. To evaluate the performance of both methods under comparable computational time constraints, we present results where QPA with one epoch is compared to LoRA with three epochs for GPT-2, and QPA with one and two epochs is compared to LoRA with five epochs for Gemma-2 (Fig. 11).
>
> The findings show that, while QPA does not outperform LoRA to the same extent when both methods are trained for the same number of epochs, it demonstrates better performance in regions with smaller parameter sizes under similar total execution times.
>
> > **Q4**. In more complex ... issues in such scenarios?
>
> **Response**: We thank the reviewer for raising this important question. To compare QPA with traditional QML methods, as discussed in Appendix A.2, it is indeed possible to generate parameters for tuning tasks using traditional QML approaches (e.g., by utilizing the expectation value of an operator as the output). However, a key distinction lies in the size of the output and the corresponding qubit requirements.
>
> For an output of size $n$, conventional QML relies on the expectation value of the Pauli-z operator, $\langle \psi (\theta) | \sigma_z^{(i)} | \psi (\theta) \rangle$, for each qubit. This approach necessitates $n$ qubits, as outlined in Eq. 14 of [3]. In contrast, QPA generates parameters by leveraging the measurement probability $|\langle \phi_i | \psi (\theta) \rangle|^2$ for each basis, which significantly reduces the qubit requirement to $\lceil \log_2 n \rceil$.
>
> In practical terms, the PEFT experiments presented in this study utilized between 4 and 11 qubits. If traditional QML were employed, the required number of qubits would scale exponentially, ranging from $2^4$ to $2^{11}$. This exponential scaling would result in an impractically large qubit requirement, rendering conventional QML infeasible for PEFT applications. Thus, in terms of scalability, the qubit usage of QPA demonstrates a significant advantage over traditional QML.

---

> ### Author Response · Authors · 2024-11-21
>
> > **Q5**. Best choice of ... hyperparameters for other models?
>
> **Response**: We thank the reviewer for providing an insightful question. The choice of block size $(n_{mlp})$ presents an interesting trade-off. Larger block sizes increase the number of trainable parameters and improve performance (lower testing perplexity), while smaller block sizes result in higher compression but require more qubits. Striking the optimal balance between these factors is likely an open question, as resource requirements vary depending on the user’s constraints and objectives.
>
> Regarding the best number of QNN repetitions ($L$) for different PEFT methods, the required qubit count can be determined once the target PEFT parameters are identified. Empirical studies suggest that $L$ typically scales proportionally with the number of qubits, either as $O(N)$ or, in some cases, $O(N^2)$ [4,5,6]. From a theoretical perspective, while studies on the overparameterization behavior of QNNs [7] indicate that QNNs can be overparameterized with a polynomial number of parameters ($O(poly(N))$), the structure of QPA differs from traditional QML approaches that use the expectation value as a loss function. Instead, QPA focuses on generating the parameters for PEFT methods. As such, the choice of $L$ within $O(N)$ or $O(N^2)$ should strike a proper balance between expressivity and avoiding overparameterization.
>
> [1] Laia Domingo, G Carlo, and F Borondo. Taking advantage of noise in quantum reservoir computing. Scientific Reports, 13(1):8790, 2023.
>
> [2] Chuhan Wu, et. al. Noisytune: A little noise can help you finetune pretrained language models better. arXiv preprint arXiv:2202.12024, 2022.
>
> [3] Andrea Mari, et al. Transfer learning in hybrid classical-quantum neural networks. Quantum, 4:340, 2020.
>
> [4] Marco Cerezo, et al. Variational quantum algorithms. Nature Reviews Physics, 3(9):625–644, 2021.
>
> [5] Sukin Sim, et al. Expressibility and entangling capability of parameterized quantum circuits for hybrid quantum-classical algorithms.  Adv. Quantum Technol., 2: 1900070
>
> [6] Marcello Benedetti, et al. Parameterized quantum circuits as machine learning models. Quantum Science and Technology, 4(4):043001, 2019.
>
> [7] Larocca, Martin, et al. "Theory of overparametrization in quantum neural networks." Nature Computational Science 3.6 (2023): 542-551.

---

> ### Author Response · Authors · 2024-11-25
> **A Kind Reminder**
>
> Dear Reviewer,
>
> With the rebuttal phase nearing its conclusion, we would like to gently remind you of our responses.
>
> We have thoughtfully addressed all your comments and questions, and we hope our revisions meet your expectations.
>
> If there are any unresolved concerns, we are more than willing to discuss them further. Otherwise, if you find our updates satisfactory, we kindly invite you to consider raising your score.
>
> Thank you again for your time and valuable feedback!

---

> > ### Comment · Reviewer_Y4nw · 2024-11-25
> >
> > Thanks for answering my questions. Q5 and Q3 exactly and perfectly addressed my questions, I would like to increase my score to a semi-positive score.

---

> > > ### Author Response · Authors · 2024-11-25
> > >
> > > We are pleased that questions Q3 and Q5 have been fully addressed. Regarding Q1, Q2, and Q4, as suggested by the reviewer, we have conducted additional experiments detailed in the appendix sections. These include examining the effects of different circuit ansatz, finite measurement shots, and noise, as well as providing a detailed comparison of QPA with traditional QML, highlighting its improved scalability. Based on these updates, we believe we have addressed these questions to a significant extent and hope the reviewer finds our responses satisfactory.
> > >
> > > We deeply value and sincerely appreciate the reviewer’s dedication and effort in providing thoughtful feedback, which has greatly contributed to improving the clarity and context of our work.

---

> > > > ### Comment · Reviewer_Y4nw · 2024-11-27
> > > >
> > > > Thanks authors for the detailed response, I need to apologize that I wasn't aware of the experiment details regarding the Q1 and Q4. I appreciate for author's efforts on addressing my and all other reviewers' concerns, based on the current version of the paper - with really interesting and reasonable idea and enough experiment details, I would like to recommend this paper appear on the ICLR this year. And I will adjust my score accordingly.

---

### Official Review · Reviewer_tsMN · 2024-11-03

**Soundness:** 3
**Presentation:** 3
**Contribution:** 3
**Rating:** 6
**Confidence:** 3

**Summary:**

The paper proposed quantum parameter adaptation (QPA), a method that combines quantum systems with classical models to enhance parameter-efficient fine-tuning in machine learning, especially for LLMs like GPT-2 and Gemma-2. Unlike traditional quantum machine learning methods that require quantum hardware for both training and inference, this approach uses quantum resources exclusively during the training phase to generate parameters for classical models, eliminating the need for quantum hardware at inference. This design enhances scalability and resource efficiency on classical hardware, addressing practical deployment concerns.

**Strengths:**

QPA achieves significant reductions in the number of trainable parameters required for fine-tuning LLMs while maintaining performance, making it highly relevant for parameter-efficient learning. This reduction makes the method resource-efficient and feasible for real-world application. In addition, the paper presents an elegant integration of quantum and classical computing. The authors demonstrate QPA’s effectiveness on state-of-the-art LLMs, contributing meaningfully to the rapidly advancing field of LLM research.

**Weaknesses:**

From my understanding, the quantum component of QPA does not appear to harness the inherent advantages of quantum machine learning. Since the quantum state must be fully characterized before classical post-processing, this process typically requires full quantum state tomography, involving an exponential number of measurements with respect to the number of qubits. This requirement is likely more computationally intensive than classical simulation, eliminating any quantum advantage. In this regard, QPA might better be described as a quantum-inspired algorithm rather than a quantum machine learning algorithm, as it does not appear to yield a speed-up or other quantum advantages over classical simulations. The authors themselves acknowledge in Appendix B that modeling the quantum state using a classical generative model could offer a more practical alternative.

**Questions:**

1.	Would the algorithm perform equally well if the quantum component were replaced with a classical system? Since the role of the parameterized quantum circuit here is to produce an intermediate hidden state, could a classical neural network serve as an alternative? Given the challenge of barren plateaus in quantum machine learning, a classical neural network might avoid these issues and potentially improve performance.
2.	Is there potential to reduce the need for full quantum state tomography in QPA? Could approaches like quantum shadow tomography, which require only partial measurement data, be used to extract sufficient information from the quantum state while reducing computational costs?
3.	The current parameterized quantum circuit design relies on CNOT and RY gates, which restrict the quantum state to real values and consequently limits the Hilbert space it explores. Have the authors experimented with more complex circuits to explore the full Hilbert space, and if so, does this yield performance improvements?

---

> ### Author Response · Authors · 2024-11-21
>
> We sincerely thank Reviewer tsMN for the valuable suggestions, which have significantly enhanced the quality of our work. We deeply appreciate the reviewer’s time and effort in providing thoughtful feedback that has helped us refine and improve our paper.
>
> > **W1**. From my understanding, the ... more practical alternative.
>
> **Response**: We appreciate the reviewer’s constructive comments. In the implementation, QPA only requires the measurement probabilities from the quantum circuit (i.e., phase information and complex amplitudes are not needed). While this is computationally intensive, advancements in fast measurement processes are both necessary and anticipated to mitigate this challenge. Furthermore, the precision of the measurement probabilities is determined by the number of measurement shots. When the learning process does not demand highly precise results (similar to the effects of quantization), it is possible to reduce the number of measurement shots required, thereby mitigating the computational burden.
>
> Although a generative model is considered one possible approach to address this issue, its role is more complementary than a full replacement. The quantum signal remains essential as the initial raw data, with the generative model assisting in enhancing or extending the data rather than entirely substituting it.
>
> From the perspective of quantum-inspired algorithms, it is feasible to leverage fast quantum simulation or approximation methods, such as tensor networks, to represent the state of a quantum circuit. However, most of these methods require circuits with limited entanglement or shallow depths to ensure that the quantum state remains simple enough to approximate. In contrast, the current proposal for the quantum circuit is designed to be more general, allowing for the construction of complex and deep quantum neural networks.
>
> > **Q1**. Would the algorithm perform ... potentially improve performance.
>
> **Response**: We express gratitude to the reviewer for asking the insightful question. We would like to clarify that the role of the quantum circuit is not only to produce an intermediate hidden state, but to produce $2^n$ distinct numbers (output measurement probabilities). For a classical neural network to achieve this, the width of the output layer should be at least $2^n$ to produce this amount of distinct outputs, for example, the shape of the final layer should be $(m, 2^n)$, where $m$ is a positive integer. This kind of substitution will obviously have more parameters than the target model or target PEFT method.
>
> For the challenge of barren plateaus raised by the reviewer, we have added a new appendix section, “On Gradient Variance of Quantum Circuit Parameters,” to address this issue. In the original study [1], barren plateaus arise from loss functions defined as the expectation value of Hermitian operators. In contrast, QPA evaluates the loss of a target classical model (an LLM in this study), with updates generated using PEFT methods. The QNN output provides measurement probabilities, not operator expectations.
>
> Our analysis of QNN gradient variance $\partial \theta$ in QPA-LoRA results for GPT-2 (Fig. 10) shows no significant downward trend with increasing qubits, likely due to backward propagation through the mapping model. A slight downward trend appears with increasing QNN repetitions $L$, resembling patterns in deep feedforward networks and barren plateaus [1,2], but to a much lesser extent. This suggests QPA’s gradient dynamics share traits of both quantum and classical paradigms without severe barren plateau effects.
>
> > **Q2**. Is there potential to ... reducing computational costs?
>
> **Response**: We appreciate the reviewer’s constructive question. While it is possible to develop a method aimed at reducing the number of quantum state measurements, however, our planned approach focuses on leveraging a pre-trained generative model to generate measurement results from a limited set of real quantum measurements.
>
> Regarding the shadow tomography method mentioned by the reviewer, its primary focus, as noted in [3], is to “recover not the full density matrix of $\rho$, but only the “shadow” that $\rho$ casts on the measurements $E_1, \ldots, E_M$.” Since this technique is designed to recover specific measurement outcomes rather than infer other measurement probabilities from known results, we currently find it less applicable to our specific challenges. However, we remain open to exploring potential methods that may prove helpful in addressing these issues as our work progresses.

---

> ### Author Response · Authors · 2024-11-21
>
> > **Q3**. The current parameterized ... performance improvements?
>
> **Response**: We sincerely appreciate the reviewer’s thoughtful question, which has helped us improve the context of our work. The rationale for using  $R_Y$  and $CNOT$ gates is rooted in the ultimate goal of generating parameters for the PEFT methods, which are typically real numbers. This reasoning guided our choice of the  $R_Y$  circuit in the main content.
>
> While it is reasonable to explore alternative circuit constructions to identify potential performance improvements, we have added Appendix F, “Effects of Different Circuit Ansatz,” in the revised version to investigate this topic. In addition to  $R_Y$  and CNOT combinations, we examined  $R_X$ + CNOT ,  $R_Y R_Z$ + CNOT , and  $R_X R_Z$ + CNOT  combinations for generating LoRA parameters in the QPA task (Fig. 7). The results indicate no significant performance differences between these circuit constructions. However, when considering a finite number of measurement shots in a noiseless environment (Fig. 9), we observed that the  $R_Y$ + CNOT combination outperforms the  $R_X$ + CNOT  combination.
>
> [1]. Jarrod R McClean, et al. Barren plateaus in quantum neural network training landscapes. Nature communications, 9(1):4812, 2018.
>
> [2]. Xavier Glorot and Yoshua Bengio. Understanding the difficulty of training deep feedforward neural networks. In Proceedings of the thirteenth international conference on artificial intelligence and statistics, pp. 249–256. JMLR Workshop and Conference Proceedings, 2010.
>
> [3] Scott Aaronson, Shadow Tomography of Quantum States, arXiv preprint arXiv:1711.01053, 2017.

---

> ### Author Response · Authors · 2024-11-25
> **A Kind Reminder**
>
> Dear Reviewer,
>
> With the rebuttal phase nearing its conclusion, we would like to gently remind you of our responses.
>
> We have thoughtfully addressed all your comments and questions, and we hope our revisions meet your expectations.
>
> If there are any unresolved concerns, we are more than willing to discuss them further. Otherwise, if you find our updates satisfactory, we kindly invite you to consider raising your score.
>
> Thank you again for your time and valuable feedback!

---

> ### Comment · Reviewer_tsMN · 2024-11-25
>
> Thank you for your detailed responses, which have addressed most of my concerns.
>
> One more follow-up question: to produce the intermediate quantum state with $2^n$ distinct amplitudes, is it necessary to perform an exponential number of measurements, or would a polynomial number suffice?
>
> I look forward to seeing your follow-up work on leveraging generative models to reduce the measurement burden.

---

> > ### Author Response · Authors · 2024-11-25
> >
> > We thank the reviewer for the thoughtful response. At the current stage, QPA utilizes a quantum circuit that produces $2^n$ measurement probabilities from each basis. Generally, if the number of measurement shots is less than $2^n$, results from some basis states may be entirely neglected due to insufficient sampling. However, if the measurement probabilities (or the quantum state) are concentrated on specific basis states, resulting in a sparse (or nearly sparse) vector representation of the probabilities, it becomes possible to accurately describe the state with fewer than $2^n$ measurement shots. In sufficiently sparse cases, a polynomial number of measurement shots may suffice. This scenario aligns with cases where the target update matrix (e.g., LoRA matrices $A$ and $B$) is sparse.
> >
> > To address the reviewer’s question, in general, the requirement for $2^n$ measurement shots holds, but in specific sparse cases, this requirement can be reduced, potentially to a polynomial scale.
> >
> > Regarding which types of PEFT learning schemes could satisfy this sparsity condition, this remains an intriguing question for future research. We deeply appreciate the reviewer for highlighting this point, as it opens up new directions and possibilities for further exploration in this area.

---

### Official Review · Reviewer_s1Tx · 2024-11-04

**Soundness:** 3
**Presentation:** 3
**Contribution:** 2
**Rating:** 6
**Confidence:** 4

**Summary:**

This paper introduces quantum parameter adaptation, a quantum parameter generation algorithm that uses quantum neural networks and classical multi-layer perceptrons to generate parameters to help fine-tune classical machine learning models.

**Strengths:**

Strengths:
   - Interesting idea (albeit one that has been done before to some extent) pushed further than in past work. Smartly gets around some of the known issues with QML, like the issue with expensive inference.
   - Brings related work closer to the forefront of classical ML by applying their method to Gemma-2 and GPT-2.
   - Good use of MLPs and batching to reduce parameter counts.

**Weaknesses:**

REVISION: The author's rebuttal and revised manuscript have addressed enough of these weaknesses to merit inclusion at ICLR.

Weaknesses:
   - Misleading claims about fault-tolerant quantum computation: The paper makes several misleading claims about fault-tolerant quantum computing (FTQC) which have the potential to make a uniformed reader think that universal FTQC is right around the corner. It’s not. For instance, the information in the second footnote is wrong. IBM has not demonstrated 12 logical qubits with 288 physical qubits. They’ve discovered a quantum error correcting code with that property. However, actually implementing that code in practice is going to be very hard! Moreover, running this paper’s parametrized quantum circuit (PQC) in a FT manner is going to be very expensive! The PQC requires small-angle Y rotations, which are extremely costly!
   - No serious discussion of the impact of noise: The paper completely sidesteps the issue of noise. This is especially concerning when the approach is outperformed by classical competitors in the noiseless setting on certain problems.
   - No theoretical performance guarantees. Not necessarily bad, but it’s not great when you don’t have amazing experimental results (the paper has no experimental results, just simulations).
   - I am not sold at all on their approach to using synthetic data to get around the issue of having finitely many shots…generative models are expensive to train too!


Some minor nitpicks:
   - It is an open question if quantum computing excels in optimization. Change the claim in your introductory paragraph, or at least soften it.
   - “This hybrid approach is especially promising in quantum machine
learning (QML), where it can enhance the training and fine-tuning of large models.” Also not evident that QML can help with the training or fine-tuning of large models. Isn’t that the whole point of this paper? Temper this claim.

**Questions:**

- Does this technique have any utility if you always need to perform a full strong simulation of the quantum circuits? Either because it is not robust to noise or because you can’t get away with finite measurements.
   - Have you tried running your method on actual quantum hardware? If not, why not?
   - How expensive would it be to implement your method in a fault-tolerant manner?
   - It is unclear to me how much this work pushes the sota forward. For instance, how does it differ from the other cited works? “Using the gradient computation and parameter update rule, the parameter generation process of QPA
has been applied to image classification with convolutional neural networks (CNNs) (Liu et al.,
2024b;a; Liu & Chen, 2024), flood prediction (time series) with long short-term memory (LSTM)
models (Lin et al., 2024), and policy gradient reinforcement learning in CartPole and MiniGrid
environments (Liu et al., 2024c).”
   - How expensive is it to update the gradient on real hardware?
   - What compute was used to produce the paper? What was the total wall clock time? How does that compare to the classical methods used?

---

> ### Author Response · Authors · 2024-11-21
>
> We sincerely thank Reviewer s1Tx for the valuable suggestions, which have greatly contributed to improving the quality of our work. We deeply appreciate the reviewer’s time and effort in helping us refine and enhance our paper.
>
> > **W1**. Misleading claims about ... which are extremely costly!
>
> **Response**: We sincerely appreciate the reviewer's thoughtful feedback and apologize for any confusion caused. The corresponding footnote has been updated to: “IBM Quantum announced the achievement of discovering a quantum error-correcting code capable of preserving up to 12 logical qubits using 288 physical qubits through error correction methods,” incorporating the reviewer’s suggestion. While real-world implementation may be challenging, we believe it is still worthwhile to explore the potential of using quantum computing techniques to fine-tune classical LLMs. This approach, which enables inference entirely on classical hardware, differs significantly from methods that aim to reconstruct transformer architectures [1] on quantum circuits. Notably, it reduces reliance on quantum computers during the inference stage.
>
> > **W2**. No serious discussion of ... noiseless setting on certain problems.
>
> **Response**: We deeply value and appreciate the reviewer’s insightful feedback. In response to the suggestion, we have added an appendix section titled “Effects of Finite Measurement Shots and Noise” in the revised version. In this section, we apply the noise model provided by IBM, derived from real hardware data from ibm_torino and ibm_fez, to investigate the potential behavior of QPA in real-device environments.
>
> Interestingly, the introduction of noise from ibm_torino and ibm_fez results in mixed outcomes: while performance slightly decreases in some cases, it improves in most instances. A similar observation has also been reported in previous quantum computing studies, where quantum noise has been found to enhance performance in specific paradigms [2].
> Similarly, in classical machine learning, studies have shown that introducing small amounts of noise can benefit the fine-tuning of LLMs [3]. Since the quantum circuit in QPA generates measurement probabilities that are fed into a classical MLP mapping model to produce PEFT parameters, it is plausible that quantum noise plays a role analogous to the “small noise” effect observed in previous studies, helping to explore more parameter spaces and aiding in the fine-tuning of LLMs.
>
> > **W3**. No theoretical performance ... results, just simulations).
>
> **Response**: We sincerely thank the reviewer for their valuable feedback and suggestions. While QPA is a newly proposed approach, theoretical analysis is planned as a future direction for further investigation. Although this study primarily focuses on simulation results, in the revised version, we have included results that incorporate a real quantum computer noise model. This addition provides insights into both ideal and more realistic scenarios.
>
> > **W4**. I am not sold ... expensive to train too!
>
> **Response**: We thank the reviewer for their valuable insights and feedback. While generative models can be challenging to train, an ideal approach to mitigate the issue of finite measurement shots could involve using a pre-trained model. This model would be trained on a general measurement result generation task and then fine-tuned for specific tasks. Nevertheless, as discussed in the appendix, we acknowledge the potential challenges associated with QPA and propose possible approaches to address them. While these suggestions may not represent definitive solutions, they aim to provide valuable insights into problem-solving paradigms for tackling such issues.
>
> > **W5**. It is an open ...or at least soften it. &
> > **W6**. “This hybrid approach ... this claim.
>
> **Response**: We appreciate the reviewer’s suggestion regarding the description in the introduction section. In response, the corresponding sentences have been revised and softened to: “Classical systems are well-suited for tasks like data processing, while quantum computing shows potential in optimization and exploring large state spaces,” and “This hybrid approach holds promise in quantum machine learning (QML), with the potential to support the training and fine-tuning of large models.”

---

> ### Author Response · Authors · 2024-11-21
>
> > **Q1**. Does this technique ... with finite measurements.
>
> **Response**: We appreciate the reviewer’s thoughtful comment and the opportunity to clarify the utility of our technique. While it is true that strong simulation of quantum circuits is required in our current implementation and situation, this does not limit the broader applicability of the proposed approach for several reasons:
>
> 1.  ***Addressing Robustness to Noise***:
> In the revised version, we have included an appendix (e.g., “Effects of Finite Measurement Shots and Noise”) where we investigate the performance of our technique under realistic noise models derived from IBM quantum hardware. Interestingly, the results show that even with noise, our method performs comparably in many cases and, in some instances, benefits from quantum noise (similar to findings in other quantum computing studies). This suggests that the technique has potential utility in noisy quantum environments, contrary to concerns about its robustness.
>
> 2. ***Finite Measurements***:
> We have also analyzed the performance of our technique with finite measurement shots. As shown in the newly added results, increasing the number of measurement shots improves performance, eventually converging toward exact simulation results. While practical implementation would require careful consideration of the trade-off between measurement counts and performance, the method remains effective even with reasonable measurement budgets. This demonstrates its potential utility beyond idealized strong simulation.
>
> 3. ***Future Practical Implementation***:
> As quantum hardware continues to improve, issues such as noise and measurement overhead are expected to become less significant. The technique’s reliance on strong simulation in its current form is primarily due to the limitations of available hardware. However, the conceptual framework and potential for generating parameter-efficient representations of PEFT methods remain valid and promising as quantum technology advances.
>
> 4. ***Hybrid Model Synergy***:
> Lastly, even when strong simulation is used, our method shows promise as a hybrid quantum-classical approach. By leveraging quantum circuits to generate low-dimensional parameter representations, the technique provides insights into integrating quantum resources with classical machine learning systems, which may inspire future noise-resilient adaptations.
>
> > **Q2**. Have you tried ... If not, why not?
>
> **Response**: At this stage, our method has not been implemented on actual quantum hardware for several reasons.  Quantum hardware at this stage is primarily shared among a large number of users, resulting in long waiting queues for access. Tasks such as fine-tuning or machine learning, which involve a large number of circuit executions, require significant computational resources and repeated runs. Obtaining results for such tasks within a reasonable timeframe becomes infeasible on current quantum hardware under shared access constraints.
>
> Given these limitations, we focused on simulations using realistic noise models derived from quantum hardware (e.g., IBM’s ibm_torino and ibm_fez). These simulations allow us to evaluate the method’s performance in noisy environments without being constrained by hardware availability or capabilities. The results, as presented in the newly added appendix section, demonstrate the robustness of our method, validating its potential for real-world quantum devices.
>
> > **Q3**. How expensive ... fault-tolerant manner?
>
> **Response**: We thank the reviewer for the valuable comments. Based on the pricing information provided by a major quantum computing provider like IBM Quantum [4], it is possible to estimate the cost of fine-tuning tasks. For instance, if the premium plan at 48 USD/minute is selected, this translates to approximately 0.8 USD/second. Assuming that a single training step involving a quantum job (including job submission, compilation, execution, and result retrieval) takes around 4 seconds [5], and considering the batch size used in our study with the WikiText-2 dataset, one epoch would require approximately 36,000 steps. The cost of executing three epochs can then be estimated as follows:
> $0.8 \times 4 \times 36,000 \times 3 = 345,600 \, \text{USD}.$
> While this cost is indeed high, as mentioned earlier, it is expected to decrease as quantum hardware improves over time. Despite these current limitations, the conceptual framework and potential for generating parameter-efficient representations remain valid and promising as quantum technology continues to advance.

---

> ### Author Response · Authors · 2024-11-21
>
> > **Q4**. It is unclear to me ... MiniGrid environments (Liu et al., 2024c).”
>
> **Response**: We thank the reviewer for this important question, as it provides an opportunity to highlight the contributions and significance of our work.
> This study pushes the state-of-the-art (SOTA) boundaries in several ways:
>
> 1. ***Model Size***:
> Previous works (as mentioned in the question) in this domain primarily focused on toy-sized models, such as a CNN model with approximately 280k parameters. While these studies are valuable, they do not adequately reflect the practical scalability or real-world applicability of the proposed methods. In contrast, our work investigates parameter-efficient learning techniques for significantly larger models, tackling up to 2 billion parameters (Gemma-2). This represents a substantial leap, moving from 280k to 2 billion parameters, and demonstrates the scalability of quantum-assisted methods when applied to LLMs.
>
> 2. ***Novel Perspective on Parameter Efficiency***:
> Instead of directly generating model parameters—a challenging and resource-intensive task—we focus on the parameters of the parameter-efficient “update” using PEFT techniques. By fine-tuning only specific layers of the model, we effectively utilize quantum resources for training while ensuring that the model remains operational entirely on classical hardware during inference. This unique perspective not only reduces the quantum hardware requirements but also makes the method more practical and applicable to large-scale systems.
>
> 3. ***First QML Application to Billion-Parameter LLMs***:
> Our work is the first in quantum machine learning (QML) to address the fine-tuning of classical LLMs in the billion-parameter range (Gemma-2), with comprehensive simulation results in both ideal and noisy situations. This achievement highlights the potential of QML to handle state-of-the-art machine learning tasks, bridging the gap between quantum and classical paradigms in a meaningful way. In contrast, most QML studies to date have focused on significantly smaller-scale tasks.
>
> In summary, this work advances the field by demonstrating the scalability and applicability of QML to real-world, large-scale models, showcasing its potential to fine-tune billion-parameter LLMs while leveraging quantum resources effectively. We believe these contributions represent a meaningful step forward in the field and hope this perspective resonates with the broader audience.
>
> > **Q5**.  How expensive is it to update the gradient on real hardware?
>
> **Response**: Similar to our response in Q3, based on the current rates, if the gradient estimation requires several circuit executions within approximately 4 seconds, the estimated cost would be around $3.2 USD per gradient update.
>
> > **Q6**. What compute was used ... classical methods used?
>
> **Response**: We sincerely appreciate the reviewer’s thoughtful question. We assume the reviewer is inquiring about the hardware used for our experiments. As detailed in Appendix C, all experiments were conducted on NVIDIA V100S and NVIDIA H100 GPUs.
>
> Regarding the total wall clock time, in the revised version, we have included execution time data in Appendix I, “On Computational Time.” Based on the LoRA results from Fig. 2, the average execution time of QPA is approximately three times longer than that of classical LoRA. This means that while QPA completes one epoch, LoRA can complete three epochs.
>
> To evaluate the performance of both methods under comparable computational time constraints, we present results where QPA with 1 epoch is compared to LoRA with 3 epochs for GPT-2, and QPA with 1 and 2 epochs is compared to LoRA with 5 epochs for Gemma-2 (Fig. 11). The findings suggest that, although QPA does not outperform LoRA to the same extent when both methods are trained for the same number of epochs, it still exhibits better performance in regions with smaller parameter sizes under similar total execution times.
>
>
> [1] Yidong Liao, Chris Ferrie, “GPT on a Quantum Computer”, arXiv preprint arXiv:2403.09418 (2024).
>
> [2] Laia Domingo, G Carlo, and F Borondo. Taking advantage of noise in quantum reservoir computing. Scientific Reports, 13(1):8790, 2023.
>
> [3] Chuhan Wu, et. al. Noisytune: A little noise can help you finetune pretrained language models better. arXiv preprint arXiv:2202.12024, 2022.
>
> [4] https://www.ibm.com/quantum/pricing
>
> [5] https://docs.quantum.ibm.com/guides/estimate-job-run-time

---

> ### Author Response · Authors · 2024-11-25
> **A Kind Reminder**
>
> Dear Reviewer,
>
> With the rebuttal phase nearing its conclusion, we would like to gently remind you of our responses.
>
> We have thoughtfully addressed all your comments and questions, and we hope our revisions meet your expectations.
>
> If there are any unresolved concerns, we are more than willing to discuss them further. Otherwise, if you find our updates satisfactory, we kindly invite you to consider raising your score.
>
> Thank you again for your time and valuable feedback!

---

> > ### Comment · Reviewer_s1Tx · 2024-11-25
> >
> > Thank you for taking the time to thoughtfully respond to my review. I appreciate the updated appendices on barren plateaus and results from simulations using fake IBM providers. It is still disappointing that there are no experimental results, which I believe to be the true test of any algorithm without an explicit fault-tolerant implementation. However, the algorithm is interesting enough, the possibility of usefulness even when relying on noiseless, strong simulations high enough, and the paper strong enough to merit inclusion at ICLR. I will revise my review appropriately.

---

> > > ### Author Response · Authors · 2024-11-25
> > >
> > > We sincerely thank the reviewer for the comment and the corresponding adjustment to the score. We would like to reiterate our heartfelt appreciation for the reviewer’s time and effort in thoroughly reviewing our paper and providing invaluable feedback.

---

### Official Review · Reviewer_jv7T · 2024-11-04

**Soundness:** 3
**Presentation:** 3
**Contribution:** 3
**Rating:** 6
**Confidence:** 3

**Summary:**

The authors propose a novel quantum circuit-based algorithm, termed Quantum Parameter Adaptation (QPA), aimed at generating parameters for classical neural networks. This approach seeks to enable parameter-efficient learning of pre-trained deep learning models by reducing the number of training parameters while preserving model performance. Specifically, they utilize a quantum neural network with tunable parameters to produce a quantum state, whose amplitude under a specific basis is processed by a classical multi-layer perceptron to generate parameters with desired properties (e.g., low rank) for fine-tuning methods. Empirical experiments are conducted on two pre-trained models, Gemma-2 and GPT-2, demonstrating that QPA achieves significant parameter reduction for fine-tuning methods like Low-Rank Adaptation (LoRA), while maintaining or improving performance in text generation tasks.

**Strengths:**

1. The authors present the Quantum Parameter Adaptation algorithm, which innovatively applies quantum devices to train classical machine learning models. This exploration opens up a new paradigm in quantum machine learning, shifting the focus from using quantum circuits solely as learning models to leveraging them for generating parameters in classical models. This shift in roles between quantum and classical computers enhances the efficiency of classical-quantum hybrid algorithms.
2. The application of QPA to practical large-scale models such as GPT-2, resulting in improved performance and parameter reduction compared to traditional classical fine-tuning methods, is commendable.

**Weaknesses:**

1. The authors assert that one motivation for proposing QPA is to mitigate the high computational costs associated with existing fine-tuning approaches by reducing the number of parameters. However, extracting amplitude information from quantum states and training the quantum neural network can be highly computationally intensive due to the inherent challenges of quantum systems and the barren plateau phenomenon. This could introduce significant computational costs during each training epoch, making a fair comparison between QPA and classical fine-tuning methods difficult.
2. Additionally, as I understand it from convergence theory [1], the number of parameters required for a quantum neural network to represent an arbitrary quantum state scales with the dimension of the quantum system, specifically $4^n$. Thus, to represent a parameterized quantum state for an arbitrary vector of dimension $r(d+k)$—the dimension of the low-rank weight matrix $\Delta W=BA$ —the quantum circuit would also need to have parameters on the order of $r(d+k)$. In this regard, QPA may not achieve any significant parameter reduction compared to completely classical fine-tuning methods.

[1]. Larocca, Martin, et al. "Theory of overparametrization in quantum neural networks." Nature Computational Science 3.6 (2023): 542-551.

**Questions:**

1. Could the authors provide experimental results comparing the performance of QPA with baseline methods while maintaining the same computational complexity?
2. Could the authors elaborate on the impact of the barren plateau phenomenon in the training of quantum neural networks, particularly concerning the optimization of large language models (LLMs) with 1 billion parameters, which necessitates a substantial number of qubits?

---

> ### Author Response · Authors · 2024-11-21
>
> We sincerely appreciate the valuable suggestions provided by Reviewer jv7T. These insights have significantly contributed to enhancing the quality of our work. We are deeply thankful for the reviewer’s time and effort in helping us refine and improve our paper.
>
> > **W1**.  The authors assert that ... classical fine-tuning methods difficult.
>
> **Response**: Thank you for the insightful comment. Reducing the number of training parameters in QPA does indeed require additional computations for quantum operations and the mapping model. As quantum computing is still in its early stages, we anticipate improvements in operation speed in the future. To compare the computational cost of QPA with classical approaches at the current stage of simulation, we have included execution time data in Appendix I, “On Computational Time.” In the LoRA results from Fig. 2, the average execution time of QPA is approximately three times longer than that of classical LoRA. This implies that while QPA completes 1 epoch, LoRA can complete 3 epochs. To assess the performance of both methods under comparable computational time constraints, we present results where QPA with 1 epoch is compared to LoRA with 3 epochs for GPT-2, and QPA with 1 and 2 epochs is compared to LoRA with 5 epochs for Gemma-2 (Fig. 11). The findings indicate that, while QPA with a similar total execution time does not outperform LoRA to the same degree as when both methods are trained for the same number of epochs, it still demonstrates better performance in regions with smaller parameter sizes.
>
> Regarding the concern about the barren plateau phenomenon, we have added Appendix H, “On Gradient Variance of Quantum Circuit Parameters” to address this topic in detail. Further elaboration on this matter will be provided in the response to Q2.
>
> > **W2**.  Additionally, as I understand it from ... classical fine-tuning methods.
>
>
> **Response**: Thank you for your constructive comments. First, regarding the reference to  $4^n$  in the work mentioned by the reviewer [1], this dimension arises from the unitary compilation task, where the goal is to decompose a target unitary  $V$  into a quantum circuit  $U(\theta)$. In this case, the Dynamical Lie Algebra (DLA) defined in that paper has a dimension of  $4^n$ . Since the task involves a quantum object as the target, such a high-dimensional requirement is understandable. However, the dimension of the DLA can vary depending on the specific task. One of the key findings of the referenced paper is that there can exist quantum neural networks (QNNs) that are overparameterized with a polynomial number of parameters.
>
> In our case, the dimension of the low-rank weight matrix is $r(d+k)$ , which approximates or represents the “update” of the weight matrix with a dimension of  $dk$  during fine-tuning. If the LoRA method is effective, this implies that the weight updates can be represented in a low-rank form. However, it is important to note that the “lower-dimensional” representation does not necessarily need to have a dimension of  $r(d+k)$. The QPA method aims to generate PEFT parameters using fewer parameters via a quantum circuit, effectively exploring a more flexible (or potentially even lower-dimensional) representation of the weight updates during fine-tuning of large language models (LLMs).
>
> Therefore, while the statement “Thus, to represent a parameterized quantum state for an arbitrary vector of dimension  $r(d+k)$ —the dimension of the low-rank weight matrix  $\Delta W = BA$ —the quantum circuit would also need to have parameters on the order of  $r(d+k)$ ” may generally be accurate, it does not fully apply to our case. In our context, the target vector (or matrix) to be represented corresponds to the weight updates during fine-tuning. Empirical results demonstrate that these updates exhibit a lower degree of freedom, allowing similar performance to be achieved with fewer training parameters.
>
> > **Q1**.  Could the authors provide ... computational complexity?
>
> **Response**: Thanks to the reviewer for giving us helpful suggestions. As we have replied in w1, we have added an appendix section of the result for comparison of QPA and LoRA with similar total execution time, and the result shows that QPA could still be beneficial in the small size parameter region.

---

> ### Author Response · Authors · 2024-11-21
>
> > **Q2**. Could the authors elaborate on the ... number of qubits?
>
> **Response**: We appreciate the reviewer’s thoughtful suggestions, which have helped us enhance the comprehensiveness of our work. Regarding the concern about the barren plateau phenomenon, we have added Appendix H, “On Gradient Variance of Quantum Circuit Parameters,” to investigate the behavior of QPA in this context.
>
> In the original barren plateau study [2], the exponential vanishing gradient behavior was derived from a learning task where the loss function is defined as the expectation value of a Hermitian operator. In contrast, our QPA approach differs. The objective function in QPA is the loss evaluated from a target classical model (an LLM in this study). The updates to this model are generated using PEFT methods, with the PEFT parameters produced by the quantum circuit and MLP mapping model. Consequently, the output of our QNN is not the expectation value of an operator but rather the measurement probabilities of basis states.
>
> To explore whether barren plateaus are present in QPA, we analyzed the variance of the QNN gradients, $\partial \theta$, in the QPA-LoRA results for GPT-2, as shown in Fig. 10. Notably, no significant downward trend in gradient variance was observed with increasing qubit counts. This behavior may be attributed to the backward propagation of QNN gradients through the subsequent mapping model, which prevents them from exhibiting the exponential vanishing trends commonly seen in expectation value-based objectives in traditional QML.
>
> Interestingly, however, a slight downward trend in gradient variance is observed as the QNN repetition $L$ increases. This behavior shows some similarity to both classical deep feedforward neural networks and barren plateaus in QNNs, as described in [2,3], albeit to a much lesser extent. This suggests that, while QPA does not exhibit barren plateaus in the same manner as traditional QML (particularly concerning qubit counts), its gradient dynamics may still reflect certain characteristics of both quantum and classical learning paradigms.
>
> Furthermore, while our largest model studied is Gemma-2, with 2 billion parameters, QPA generates only the PEFT parameters (e.g., LoRA parameters) instead of directly generating the full model parameters. This significantly reduces the required number of qubits compared to directly handling the full model. As a result, the maximum number of qubits used in this study is limited to 11.
>
>
> [1] Larocca, Martin, et al. "Theory of overparametrization in quantum neural networks." Nature Computational Science 3.6 (2023): 542-551.
>
> [2] Jarrod R McClean, et al. Barren plateaus in quantum neural network training landscapes. Nature communications, 9(1):4812, 2018.
>
> [3] Xavier Glorot and Yoshua Bengio. Understanding the difficulty of training deep feedforward neural networks. In Proceedings of the thirteenth international conference on artificial intelligence and statistics, pp. 249–256. JMLR Workshop and Conference Proceedings, 2010.

---

> > ### Comment · Reviewer_jv7T · 2024-11-26
> >
> > Thanks for the author's detailed response, which has partially addressed my concerns. However, I have two further questions based on the response:
> > 1. Regarding the additional numerical results that consider computational time, the authors note that "while QPA with a similar total execution time does not outperform LoRA to the same degree as when both methods are trained for the same number of epochs, it still demonstrates better performance in regions with smaller parameter sizes." Could the reliance on small parameter sizes, where QPA outperforms baseline models, limit the practical applicability of QPA, given that many real-world models involve a large number of parameters?
> >  2. Regarding the response to the barren plateau problem, the authors state that "the output of our QNN is not the expectation value of an operator but rather the measurement probabilities of basis states." However, aren't the measurement probabilities of basis states mathematically related to the expectation values of corresponding measurement operators? Furthermore, the authors consider the measurement probabilities of all basis states, which would entail an exponential computational cost as the number of qubits increases.

---

> > > ### Author Response · Authors · 2024-11-27
> > >
> > > We thank the reviewer for their thoughtful questions.
> > >
> > > > Regarding the ... large number of parameters?
> > >
> > > Firstly, the term “smaller parameter size” refers to scenarios where fewer training parameters are used when fine-tuning the same model. In other words, QPA demonstrates superior performance in cases of extreme compression. For instance, in real-world applications, we can select the desired level of compression during fine-tuning. In the “smaller parameter size” region of PEFT methods, QPA outperforms its counterparts. We hope this clarification addresses the reviewer’s question.
> > >
> > > > Regarding the response to the barren plateau ... measurement operators?
> > >
> > > Regarding the calculation of the expectation value of an operator, yes, it can be derived from the probabilities generated by quantum states. However, as explained in our earlier response, QPA does not terminate at the QNN’s output (unlike traditional QML, which directly uses the expectation value as the objective). Instead, QPA includes additional processes to generate the loss function as the final objective.
> > >
> > > > Furthermore, the authors ... number of qubits increases.
> > >
> > > Finally, while the Hilbert space grows exponentially with the number of qubits, in our case, the required qubit count is logarithmic with respect to the number of parameters in the target PEFT method. This allows us to effectively utilize the exponential space, aligning it with the original scale of PEFT parameters, by controlling the QNN with the corresponding number of qubits.
> > >
> > > Once again, we sincerely appreciate the reviewer’s time and effort in thoroughly reviewing our paper and providing invaluable feedback. In response to the reviewer’s comments, we have added additional appendix sections to address questions regarding both the comparison while maintaining the same computational complexity and the investigation of the effect of barren plateaus in QPA. If the reviewer find our updates satisfactory, we kindly invite the reviewer to consider raising your score.
> > > Thank you again for your time and valuable feedback!

---

> ### Author Response · Authors · 2024-11-25
> **A Kind Reminder**
>
> Dear Reviewer,
>
> With the rebuttal phase nearing its conclusion, we would like to gently remind you of our responses.
>
> We have thoughtfully addressed all your comments and questions, and we hope our revisions meet your expectations.
>
> If there are any unresolved concerns, we are more than willing to discuss them further. Otherwise, if you find our updates satisfactory, we kindly invite you to consider raising your score.
>
> Thank you again for your time and valuable feedback!

---

### Author Response · Authors · 2024-11-21
**Global Response**

Dear Reviewers,

We thank the reviewers for their insightful questions and feedback. We have carefully addressed all the comments and provided detailed responses in the revision and rebuttal. The latest revision is now available, with all changes and modifications from the original submission highlighted in brown for clarity.

In response to the reviewers’ suggestions, we have made several updates to improve the completeness of the paper. These include new appendix sections on pages 18–22, discussing the effects of finite measurement shots and noise, examining different circuit ansatz, and investigating the impact of barren plateaus.

---
## Revision Details (please also see the revised PDF)

Major revisions include:
* New sections added:
    * Appendix F: Effects of Different Circuit Ansatz [ tsMN, Y4nw ]
    * Appendix G: Effects of Finite Measurement Shots and Noise [ s1Tx , Y4nw , KJS2 ]
    * Appendix H: On Gradient Variance of Quantum Circuit Parameters [ jv7T , tsMN , KJS2 ]
    * Appendix I: On Computational Time [ jv7T , s1Tx , Y4nw ]

Minor revisions include:
* Enhance the introduction and refine the description of IBM’s error correction achievement [ s1Tx ]
* Updated the main text to include references to the relevant appendix sections.
* Correct typos.

We hope these revisions effectively address the reviewers’ concerns and enhance the overall quality of our paper.

Thank you once again for your valuable feedback!

---

### Meta-Review · Area_Chair_x7da · 2024-12-20

**Metareview:**

This paper presents Quantum Parameter Adaptation (QPA), a novel framework for quantum parameter generation aimed at enhancing parameter-efficient fine-tuning of large language models (LLMs). By leveraging quantum neural networks (QNNs) during training, the proposed approach generates parameters for classical models, enabling inference exclusively on classical hardware and addressing key challenges related to quantum resource demands. Simulation results highlight QPA's ability to reduce the number of parameters while maintaining or slightly improving performance on text generation tasks, as demonstrated with GPT-2 and Gemma-2 case studies.

Reviewers acknowledged the innovative integration of quantum-classical hybrid techniques and the empirical validation of baseline LLMs. However, they also expressed concerns regarding computational efficiency, scalability, and the lack of theoretical guarantees for QPA's performance. Despite these limitations, the paper makes certain contributions to the development of hybrid quantum-classical learning frameworks.

**Additional Comments On Reviewer Discussion:**

During the rebuttal period, the authors and reviewers engaged in discussions on key issues, including the computational efficiency of QPA, its scalability to larger LLMs, and the lack of theoretical guarantees for its performance. Reviewers also questioned the necessity of QNNs over classical alternatives and expressed doubts about the practical implications of relying on simulated quantum results without hardware implementation.

In response, the authors provided empirical results and added appendix sections addressing scalability, noise resilience, and the effects of finite measurement shots. They clarified the role of QNNs in generating parameter-efficient representations, drew comparisons with tensor network alternatives, and expanded the discussion on trade-offs with classical methods. While concerns about hardware feasibility and computational trade-offs remain, the authors' revisions and responses greatly improved the paper's clarity.

---

### Decision · Program_Chairs · 2025-01-22

Accept (Poster)